# Spectral signature and behavioral consequence of spontaneous shifts of pupil-linked arousal in human

**Ella Podvalny[1], Leana E King[1], Biyu J He[1,2]\***

[1]Neuroscience Institute, New York University School of Medicine, New York, United States; [2]Departments of Neurology, Neuroscience & Physiology, and Radiology, New York University School of Medicine, New York, United States

**Abstract** Arousal levels perpetually rise and fall spontaneously. How markers of arousal—pupil size and frequency content of brain activity—relate to each other and influence behavior in humans is poorly understood. We simultaneously monitored magnetoencephalography and pupil in healthy volunteers at rest and during a visual perceptual decision-making task. Spontaneously varying pupil size correlates with power of brain activity in most frequency bands across large-scale resting state cortical networks. Pupil size recorded at prestimulus baseline correlates with subsequent shifts in detection bias (*c*) and sensitivity (*d'*). When dissociated from pupil-linked state, prestimulus spectral power of resting state networks still predicts perceptual behavior. Fast spontaneous pupil constriction and dilation correlate with large-scale brain activity as well but not perceptual behavior. Our results illuminate the relation between central and peripheral arousal markers and their respective roles in human perceptual decision-making.

## Introduction

Arousal level constantly fluctuates during wakefulness. These spontaneous fluctuations are controlled by brain states that constrain sensory information processing and shape behavioral responses (*McCormick et al., 2020*; *Joshi and Gold, 2020*). To infer the level of arousal, investigators typically use one of two physiological markers: pupil size and spectral content of cortical activity. Larger pupil or more desynchronized (i.e., higher-frequency) brain activity generally correspond to higher arousal. While the two arousal markers correlate with each other in animals (*Stitt et al., 2018*; *McGinley et al., 2015b*; *Joshi et al., 2016*; *Yüzgeç et al., 2018*; *McGinley et al., 2015a*; *Reimer et al., 2014*), they historically have motivated two separate lines of research in humans (outlined below). In addition, it remains unknown to what extent spectral content of cortical activity serves functional roles distinct from those related to pupil-linked arousal. A lack of understanding on how the two physiological markers of arousal relate to each other in humans prevents the synthesis of previous findings and leaves new findings in humans difficult to interpret in the context of the quickly advancing research of arousal in animal models.

The quest to characterize the behavioral states associated with changes in pupil size in humans began more than a century ago. The Yerkes–Dodson law (1908) (*Yerkes and Dodson, 1908*) describes behavioral performance in difficult tasks as an inverted-U function of arousal, where intermediate levels of arousal (inferred from pupil size) are most beneficial to task performance. This law has been criticized, however, for not taking into account that performance efficiency cannot be captured by a single behavioral variable in most tasks (*Eysenck, 1982*) and later studies have focused on more extensive research of pupil-linked behavior (e.g., *Kahneman and Beatty, 1966*). These studies, however, did not make the distinction between ongoing pupil-linked state (e.g., prestimulus baseline)

**\*For correspondence:**
biyu.jade.he@gmail.com

**Competing interest:** The authors declare that no competing interests exist.

and event-related pupillary response (*Goldwater, 1972*) that are likely governed by distinct neuro-modulatory circuits and/or modes of brain activity (*McCormick et al., 2020*; *Joshi and Gold, 2020*; *Aston-Jones and Cohen, 2005*). A few recent studies show that baseline pupil size predicts perceptual decisions in humans (*Waschke et al., 2019*; *van Kempen et al., 2019*) and animals (*McGinley et al., 2015a*; *Steinmetz et al., 2019*), but the timescale of the arousal fluctuations shaping behavior and their underlying cortical mechanisms remain poorly understood.

A parallel line of research concerns the spectral content of electrophysiological activity and its relationship to behavior. At a very slow timescale, such as during a transition between states of waking and sleep, variations in the broadband power spectrum of cortical EEG activity are well documented in humans and animals (e.g., *Buzsáki, 2006*; *He et al., 2010*). More recent research reveals that changes in cortical spectral power within the waking state (e.g., measured as prestimulus baseline spectral power) also predict (trial-to-trial) variations in behavior. At this timescale, while some animal studies show that broadband changes in cortical spectral power correlate with moment-to-moment arousal fluctuation and influence behavioral performance (*McGinley et al., 2015b*; *Sederberg et al., 2019*), human studies mostly focus on band-limited power with no control for spontaneous arousal shifts (*Linkenkaer-Hansen et al., 2004*; *Wyart and Tallon-Baudry, 2009*; *Arnal et al., 2015*). For example, prestimulus power in the alpha band (*Samaha et al., 2020*) inversely relates to stimulus detection (*Ergenoglu et al., 2004*; *van Dijk et al., 2008*; *Limbach and Corballis, 2016*; *Busch et al., 2009*) and attentional allocation (*Jensen et al., 2012*). Whether the effect of prestimulus power in alpha and other frequency bands on behavioral performance can be partially explained by spontaneous fluctuations of arousal remains unknown (*van Kempen et al., 2019*; *Hong et al., 2014*).

Importantly, previous studies in this domain have not investigated how pupil-linked arousal covaries with large-scale cortical spectral power across a wide range of frequencies. Without understanding the respective contributions of general arousal states and arousal-independent changes in cortical activity, a comprehensive view of the neural mechanisms governing state-dependent behavior remains elusive. Supporting this distinction, a recent study (*Waschke et al., 2019*) showed that pupil-linked arousal and desynchronization in auditory cortex have distinct contributions to perceptual behavior, but this study only investigated the auditory cortex.

The overarching goal of our study is to shed light on how ongoing fluctuations in brain state shape human visual perceptual decision-making, using a task that probes both detection and discrimination, and metrics that capture perceptual sensitivity, criterion, and reaction time. Using simultaneous recordings of brain activity (magnetoencephalography [MEG]) and pupil size in resting state and in task, we uncover a strong association between pupil size and large-scale cortical power across a wide range of frequencies. MEG provides superior signal-to-noise ratio and source modeling allowed us to investigate the unique behavioral contributions of spectral power modulation in different large-scale brain networks. We show that pupil size fluctuations reflect antagonistic shifts in spectral power between low-frequency and high-frequency bands, while the intermediate alpha band relates to pupil size according to an inverted-U function. Both pupil-linked and pupil-independent brain states influence perceptual decision-making, albeit in different manners. These findings indicate that the prestimulus brain state influencing perceptual decisions contains a strong component linked to global electrophysiological arousal fluctuations in conjunction with the unique contributions of specific frequency bands localized to specialized brain networks.

## Results

We simultaneously monitored pupil size and brain activity (using MEG) in 24 participants during eyes-open rest and a visual perceptual decision-making task (*Figure 1A–B*). During eyes-open rest, participants were instructed to fixate on a central fixation cross and avoid any focused mental activity. We acquired two 5 min rest sessions in 21 participants and one rest session in the three remaining participants (see Materials and methods and *Figure 1B*). The task included long prestimulus intervals (3–6 s) wherein a fixation cross on a gray background was identical to the one used in eyes-open rest sessions (*Figure 1C*). This design allowed us to investigate the spontaneous variation in pupil size and brain activity under identical and minimal visual stimulation during the eyes-open rest and the prestimulus task baseline.

Pupil size was monitored at all times during the experiment. *Figure 1D* depicts a 1-min example of such recording during rest before (light gray) and after (dark gray) exclusion of blink periods (see

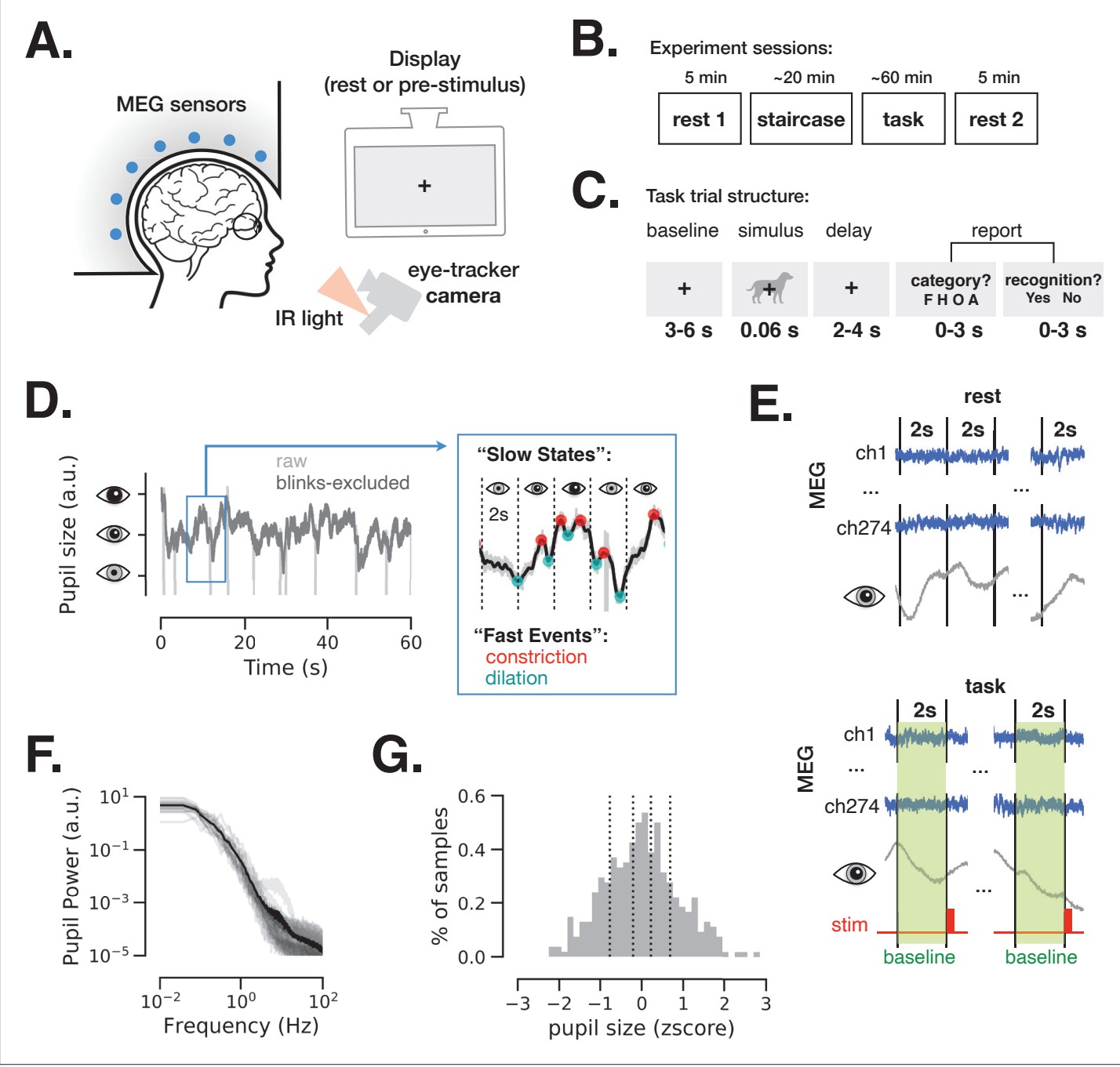

**Figure 1.** Magnetoencephalography (MEG) and pupil size monitoring in rest and task. (**A**) The experiment included simultaneous MEG recordings and eye-tracking while participants (*N* = 24) fixated at all times the fixation cross is present. (**B**) Acquisition timeline: two 5 min rest sessions (beginning and end), a staircase procedure determining the threshold contrast and a main task involving liminal stimuli presented at a staircase-determined contrast. (**C**) Each trial of the task included a prestimulus interval, followed by a liminal object stimulus (i.e., leading to ~50 % 'yes' reports) and two forced-choice decisions: first, 'category' (face, house, object, or animal) of the stimulus and, second, 'recognition' (yes or no) to indicate whether a meaningful stimulus was perceived or not. A meaningful stimulus was present on most trials (n = 300) whereas a scrambled image was presented on the remaining trials (n = 60, more details in **Figure 4**). (**D**) An example 1-min pupil size recording during rest. Light gray trace depicts raw data and dark gray trace depicts the same time course with blink periods excluded and 5 Hz low-pass filter applied. The blue-frame inset shows magnified 10 s recording with pupillary constriction/dilation 'events' (red/cyan) and examples of slower 'states' spanning 2 s non-overlapping time windows. (**E**) To study variation in slow states, we defined consecutive non-overlapping 2 s epochs in rest and 2 s baseline periods before each stimulus presentation in task. (**F**) Power spectrum of a 5 min pupil size recording in rest reveals aperiodic pupil fluctuations since no oscillatory peaks are evident. Transparent light gray curves denote individual subjects and solid black curve denotes across-subject average spectrum. (**G**) Example distribution of 'slow state' pupil size (i.e., averaged in

*Figure 1 continued on next page*

*Figure 1 continued*
2 s windows) recorded during rest for one subject. The black lines depict percentiles (20, 40, 60, 80) according to which the 2 s windows were split in groups in Figures 2 and 4. Source data is available as a supplementary file.

The online version of this article includes the following source data for figure 1:

**Source data 1.** Source data for *Figure 1F*, including power spectrum of pupil size fluctuations from individual subjects.

Materials and methods). The pupil size was analyzed according to 'slow states' and 'fast events', capturing two timescales of spontaneous pupil changes. 'Slow states' are defined as the averaged pupil size within 2 s non-overlapping windows (*Figure 1E*). The time interval of 2 s considers the slow nature of pupil size fluctuations, as shown in *Figure 1F*: the power spectrum of pupil size fluctuations is dominated by power in low frequencies, exhibiting a 1/f-type spectrum, consistent with previous reports (*Yellin et al., 2015*). The longer timescale (2 s windows) was also chosen to facilitate spectral analysis of brain activity at low-frequency ranges (delta-theta). Pupil size extracted from 2 s windows was approximately normally distributed (*Figure 1G*). 'Fast events' are defined as spontaneous momentary constrictions and dilations, starting at local peaks and troughs in the pupil size timeseries, respectively (*Figure 1D*, blue-frame inset).

While baseline pupil diameter ('slow state' equivalent) correlates with subsequent perceptual detection in humans (*Waschke et al., 2019*; *van Kempen et al., 2019*; *Podvalny et al., 2019*) and mice (*McGinley et al., 2015a, Aston-Jones and Cohen, 2005*; *Steinmetz et al., 2019*), it is currently unknown whether momentary spontaneous dilations and constriction ('fast events') during a baseline period play a functional role in behavior. Animal studies show that fast pupil events are regulated by neural activity in cortex, superior colliculi, and locus coeruleus (LC) (*Joshi et al., 2016*; *Reimer et al., 2014*; *Reimer et al., 2016*), with distinct neuromodulatory control as compared to slow pupil states: rapid and longer-lasting pupil dilations are associated with phasic activity of LC noradrenergic (LC-NE) projections and sustained activity in basal forebrain cholinergic (BF-Ach) projections, respectively (*Reimer et al., 2016*). In the following sections, we report the electrophysiological neural correlates of pupillary slow states and fast events and expose the behavioral consequences of these spontaneous changes.

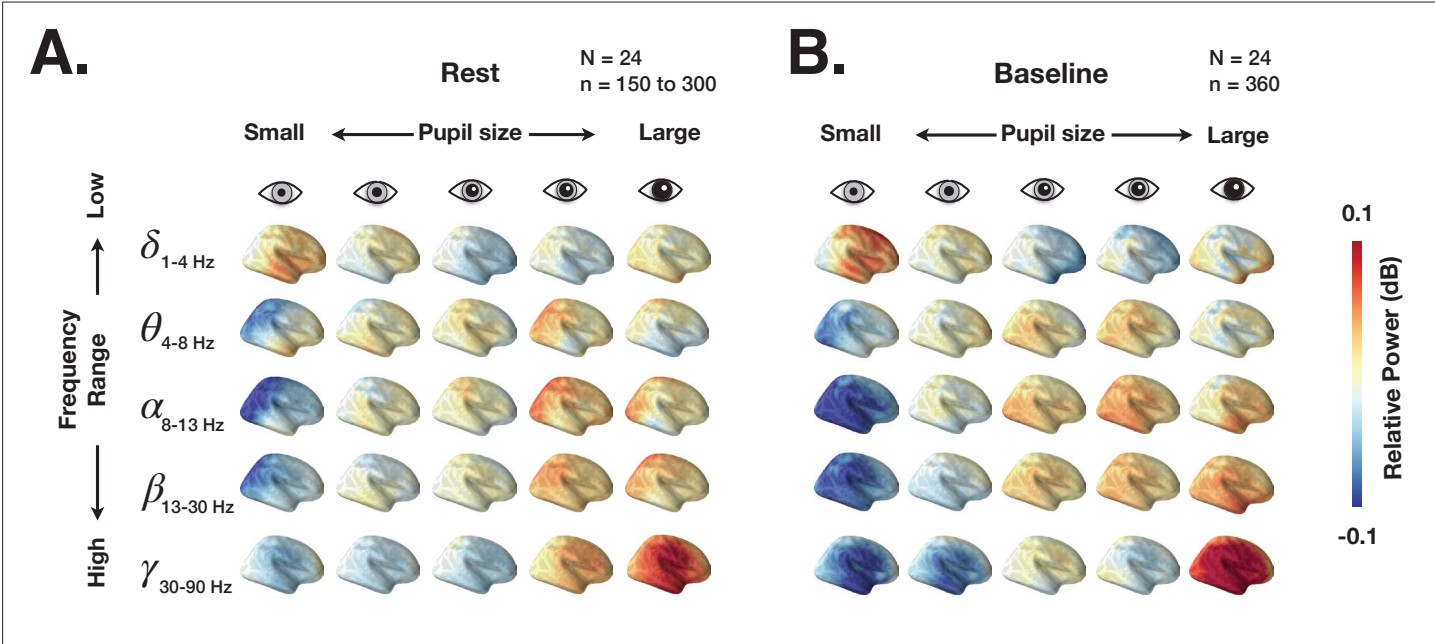

**Figure 2.** Dynamic imaging of coherent sources (DICS) power maps of resting state and prestimulus baseline brain activity grouped according to pupil size. (**A**) The source power analysis (DICS) conducted in 2 s non-overlapping time windows recorded during continuous resting state fixation. The 2 s windows were grouped according to the five pupil size percentiles (0–20%, 20–40%, 40–60%, 60–80%, 80–100%). Power in each frequency band is normalized by average power across all pupil sizes. (**B**) Same as A but for 2 s prestimulus intervals extracted from task data.

# Spectral power covaries with pupil size across large-scale cortical networks

We first investigate the link between spectral content of ongoing brain activity and pupil size during eyes-open rest and prestimulus baseline in task. The spectral power of brain activity in 2 s epochs was localized using dynamic imaging of coherent sources (DICS) (*Gross et al., 2001*) (for details, see Materials and methods). *Figure 2* depicts the distribution of power across the whole brain grouped by 'slow state' pupil size (using percentiles illustrated in *Figure 1G*). These results show widely distributed changes in power with slow fluctuations of pupil size: low-frequency (delta) power decreases and high-frequency (beta-gamma) power increases with larger pupil size. Such widespread cortical activity modulation could stem from the activity of neuromodulatory systems that regulate pupil size, such as the LC system, with LC-NE axons projecting widely to many cortical areas (*Aston-Jones and Cohen, 2005*).

We next subjected the single-trial DICS power maps to quantitative analyses according to large-scale brain networks. Because human (*Yellin et al., 2015*) and monkey (*Chang et al., 2016*) functional magnetic resonance imaging (fMRI) studies report correlations between pupil size and activity in large-scale resting state networks (RSNs), we used a functional atlas (*Yeo et al., 2011*) estimated from a large resting state fMRI dataset (*N* = 1000) by clustering functionally connected cortical areas into seven large-scale RSNs (*Figure 3A*). Previous work using resting state MEG data have also revealed similar topography of large-scale networks (*de Pasquale et al., 2010*; *Hipp et al., 2012*).

The relationship between pupil size and spectral power was investigated using a linear mixed-effects model (LMM, see Materials and methods). For each frequency band, we averaged spectral power within each RSN (colored areas in *Figure 3A*) and fit an LMM to samples of individual 2 s windows of rest and task baseline (which avoids the arbitrary percentile grouping). We fit two types of models, one with both linear and quadratic components (*Equation 3*, Materials and methods), following previous reports of the relationship between pupil size and membrane potentials in the mouse auditory cortex (*McGinley et al., 2015a*), and one with a linear component only (*Equation 4*, Materials and methods) for comparison. *Figure 3B* presents the fitted curves from the task baseline data (similar plots from rest are shown in *Figure 3—figure supplement 1*) for the models with lower Bayesian information criterion (BIC), and *Figure 3C* presents the linear parameter estimates for both task baseline and rest, and a quadratic parameter estimate in case the model that included this parameter was preferred according to BIC. The parameter estimates vary across frequency bands but follow a qualitatively similar pattern across RSNs, reinforcing the above impression of widespread pupil-linked spectral power changes (*Figure 2*). The full statistics are given in *Table 1*. Below, we describe the results of individual frequency bands in detail.

Delta power generally decreases with increasing pupil size, as can be inferred from the negative parameter estimate of the linear model component ($\chi_L < 0$; *Figure 3C*, right), which was significant in somatomotor, ventral attention, frontoparietal, and default mode networks in the prestimulus baseline, but not at rest (FDR-corrected). This finding is consistent with previous studies reporting a reduction in delta power of cortical EEG and of membrane potential with increased arousal, such as during walking (*McGinley et al., 2015a*; *Buzsaki et al., 1988*). It is also consistent with findings of higher delta power during deep sleep, corresponding to a very low arousal state. A positive quadratic component ($\chi_Q > 0$) can be observed in the limbic network, for both baseline and rest data, suggesting a more complex U-shaped relationship (*Figure 3C*, left).

Theta power shows an inverted U-shaped relationship with pupil size ($\chi_Q < 0$), which was significant at both task baseline and rest in dorsal attention network (DAN) and at baseline in visual network. The linear component was mostly positive ($\chi_L > 0$), indicating that theta power increases with larger pupil size, but this effect was not significant in the RSNs we examined.

The relationship between alpha power and pupil size had a negative quadratic component ($\chi_Q < 0$) in all RSNs except the limbic network at both rest and task baseline where the model with the quadratic component produced a lower BIC, indicating an inverted U-shaped relationship. This finding is consistent with a previous EEG study (*Hong et al., 2014*). The linear component was positive in all RSNs and both conditions, pointing to a nonintuitive trend of alpha power increasing with higher pupil-linked arousal. Consistent with recent suggestions (*Sadaghiani and Kleinschmidt, 2016*; *Clayton et al., 2018*), these results indicate that eyes-open alpha power does not index transitioning to a state of relaxation ('idling') as was traditionally considered in EEG research.

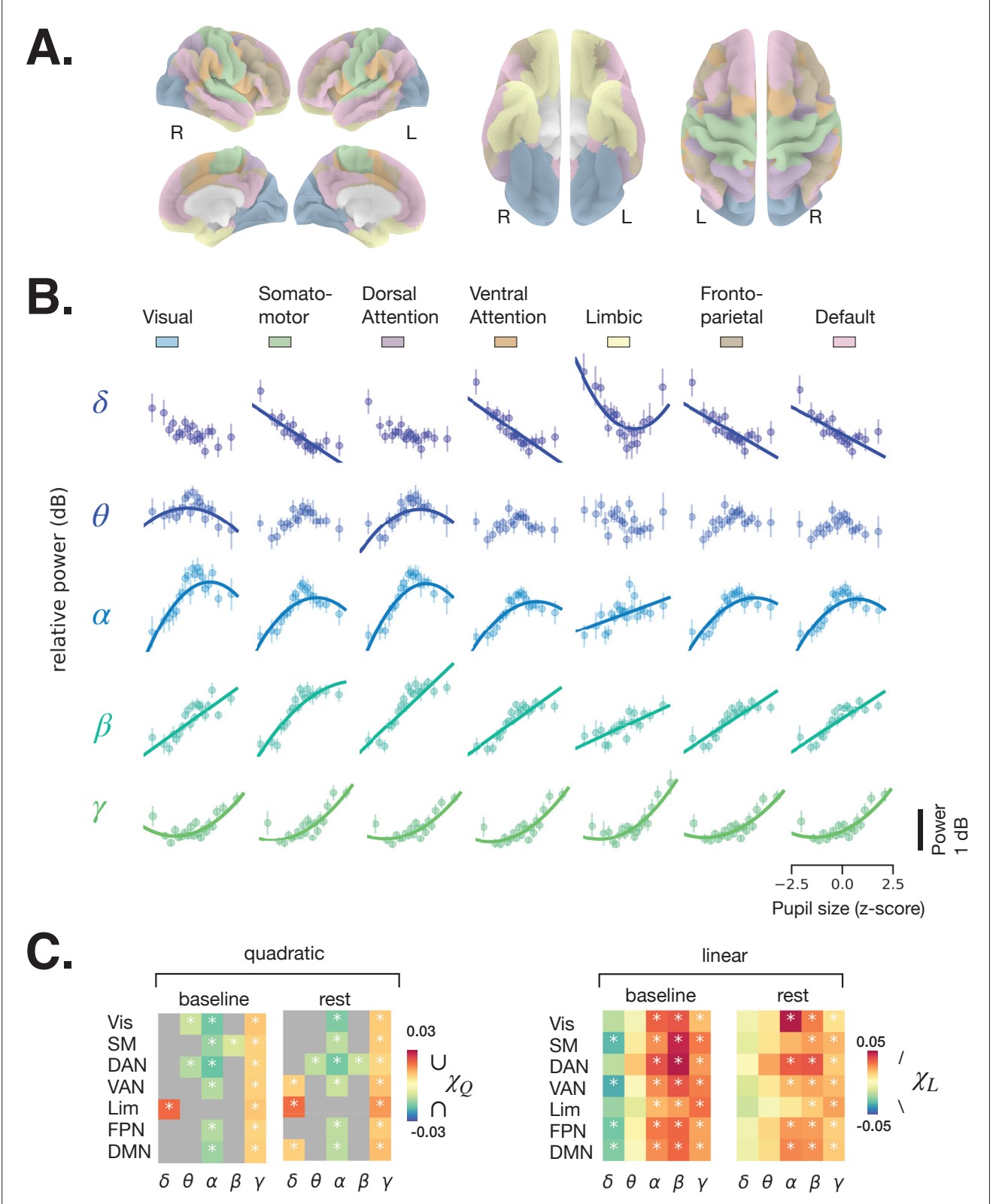

**Figure 3.** Spectral power of spontaneous activity in all frequency bands covaries with pupil size across large-scale resting state networks (RSNs). (**A**) Human functional RSNs according to Yeo 2011 atlas (*Yeo et al., 2011*). (**B**) Predicted fit of spectral power based on pupil size was estimated using linear mixed models in prestimulus baseline data (equivalent results from rest are shown in *Figure 3—figure supplement 1*, and single-subject data in *Figure 3—figure supplement 2* and *Figure 3—figure supplement 3*). For visualization purposes only, the spectral power was binned by pupil size

*Figure 3 continued on next page*

*Figure 3 continued*

percentiles in 5 % intervals; colored circles and error bars represent mean and standard error of the mean across subjects, respectively. The models were fit using all individual epochs (n = 8460 prestimulus, n = 6705 rest, N = 24). Only models with a lower Bayesian information criterion (BIC) and a significant parameter estimate (p < 0.05, FDR-corrected) are plotted. (**C**) Parameter estimates of the quadratic ($\chi_Q$) and linear ($\chi_L$) components of each model fit to resting state and prestimulus baseline data. Gray color indicates that a model with a linear component only was preferred (lower BIC), otherwise both $\chi_Q$ and $\chi_L$ are plotted for the quadratic model. Vis: visual, SM: somatomotor, DAN: dorsal attention network, VAN: ventral attention network, Lim: limbic, FPN: frontoparietal network, DMN: default-mode network. Asterisks denote p < 0.05 after FDR correction across RSNs. Figure source data is available as a supplementary file.

The online version of this article includes the following source data and figure supplement(s) for figure 3:

**Source data 1.** Source data for *Figure 3B*, including individual subject-level data for power-pupil relationship per RSN and frequency band using prestimulus baseline data.

**Source data 2.** Source data for *Figure 3—figure supplement 1*, including individual subject-level data for power-pupil relationship per RSN and frequency band using resting state data.

**Figure supplement 1.** Same as *Figure 3B*, but for resting state data.

**Figure supplement 2.** Same as *Figure 3B*, with individual subject data.

**Figure supplement 3.** Same as *Figure 3B*, but for resting state with individual subject data.

**Figure supplement 4.** Analysis of pupil-linked spectral power in sensor-level magnetoencephalography (MEG) data.

Both beta and gamma power show strong positive linear correlation ($\chi_L > 0$) with pupil size in all RSNs and both conditions. Such a positive correlation was recently observed in the EEG beta range, but not in the gamma range (*Waschke et al., 2019*). The inconsistency in the gamma range may stem from the superior MEG sensitivity to high-frequency activity. In addition, beta and gamma power exhibit weakly negative and positive quadratic components, respectively (*Figure 3C*, left), consistent with their respective saturating and U-shaped relationship with pupil size (*Figure 3B*).

While source modeling was necessary to determine how cortical power changes with pupil size, we tested whether a simpler spectral analysis in the sensor space leads to comparable results. To this end, we estimated the power spectrum for each MEG sensor in every 2 s epoch and fit a mixed-effects model for each sensor to estimate the linear and quadratic components according to pupil size

**Table 1.** Behavioral consequence of pupil-linked states.

Gray shading indicates models plotted in Figure 4C–D, where p < 0.05 for $\chi_Q$ in Q-models and $\chi_L$ in L-models. HR: hit rate, FAR: false alarm rate; *c*: criterion, *d′*: sensitivity, Acc: accuracy, RT: reaction time. L-model and Q-model denote models with linear only or both linear and quadratic components, specified in and *Equations 4* and *5*, respectively. Marginal and conditional $R^2$ indicate the proportion of total variance explained by fixed effects only and the proportion of variance explained by both fixed and random effects, respectively. BIC: Bayesian information criterion.

| | **BHV** | $\chi_Q$ | Std | pval | $\chi_L$ | Std | pval | BIC | Marginal $R^2$ | Conditional $R^2$ |
|---|---|---|---|---|---|---|---|---|---|---|
| Q-model | HR | −0.020 | 0.007 | 0.006 | 0.043 | 0.007 | 8.48E-09 | −166.659 | 0.037 | 0.923 |
| | FAR | 0.002 | 0.010 | 0.85 | 0.009 | 0.009 | 0.299 | −131.423 | 0.002 | 0.850 |
| | *c* | 0.021 | 0.022 | 0.34 | −0.083 | 0.022 | 1.655E-04 | 82.371 | 0.016 | 0.917 |
| | *d′* | −0.080 | 0.034 | 0.0196 | 0.098 | 0.017 | 4.38E-09 | 141.827 | 0.036 | 0.660 |
| | Acc | −0.013 | 0.007 | 0.063 | 0.031 | 0.008 | 3.745E-05 | −205.201 | 0.052 | 0.816 |
| | RT | 0.013 | 0.008 | 0.096 | −0.029 | 0.008 | 7.248E-04 | −192.939 | 0.020 | 0.938 |
| L-model | HR | | | | 0.043 | 0.007 | 6.943E−09 | −178.111 | 0.057 | 0.846 |
| | FAR | | | | 0.009 | 0.009 | 0.299 | −149.132 | 0.003 | 0.746 |
| | *c* | | | | −0.083 | 0.022 | 1.31E−04 | 67.171 | 0.027 | 0.848 |
| | *d′* | | | | 0.098 | 0.039 | 0.011 | 187.050 | 0.036 | 0.446 |
| | Acc | | | | 0.031 | 0.008 | 3.539E−05 | −220.347 | 0.078 | 0.676 |
| | RT | | | | −0.029 | 0.008 | 7.346E−04 | −200.883 | 0.033 | 0.854 |

(*Figure 3—figure supplement 4*). The fitted parameter estimates of the models in the sensor space were consistent with those from the source-space analysis.

Together, these findings indicate that power in most large-scale networks and most frequency bands significantly correlates with pupil size according to at least one model component (linear, $\chi_L$, and/or quadratic, $\chi_Q$) in models fit to prestimulus baseline data. While the parameter estimates were also generally similar between the baseline and rest models, several rest models did not show a significant coupling between power and pupil size. This discrepancy could either stem from suboptimal model fit in rest (where less data is available) or from decoupling of activity in some RSNs from the neuromodulatory systems governing arousal during rest.

## Behavioral consequence of prestimulus pupil-linked states

Next, we investigated how spontaneous fluctuations in pupil-linked brain states shape perceptual decisions in an object recognition task. The task details and behavior were previously reported (*Podvalny et al., 2019*). Briefly, the task entails recognition and categorization of liminal object images (*Figure 1C*): the image contrast was titrated for each participant in a pre-task staircase procedure, such that the same image was reported as recognized in 44.9% ± 3.5% (i.e., %'yes' reports, mean ± s.e.m., *N* = 24) of trials during the main task, not significantly different from the intended 50 % recognition rate. The task included real images and their scrambled counterparts. Scrambled images were generated by phase-scrambling real images, which preserves low-level image features that differ between categories. Real-image trials were used to determine the hit rate and scrambled-image trials were used to determine the false alarm rate (*Figure 4A*), which were subsequently used to calculate shifts in criterion (*c*) and sensitivity (*d'*) as a function of prestimulus pupil/brain state (*Figure 4B*). In addition to these subjective recognition reports ('yes'/'no'), subjects were also asked to report the object category ('face'/'animal'/'house'/'object') whereby they were instructed to guess the category if they could not recognize an object. Categorization accuracy (%correct) and reaction time were obtained from the category reports for both real and scrambled images.

To investigate how perceptual behavior is affected by pupil-linked arousal, we sorted task trials into five groups (with equal number of trials) according to the prestimulus pupil size ('slow states') and calculated behavioral metrics within each group of trials (see Materials and methods and *Figure 4A*). Previous studies reported linear and/or quadric models fitting such behavioral metrics and task-evoked pupil responses (*McGinley et al., 2015a*; *Waschke et al., 2019*; *de Gee et al., 2017*) and, accordingly, we also report the results of both model types to allow comparison (see Materials and methods, *Equations 5–6*).

Our results indicate that hit rate (i.e., proportion of 'yes' responses to images containing real objects) increases with baseline pupil size (*Figure 4C*), whereby both model types produce significant positive linear component (p < 0.05, see *Table 1* for exact statistics). By contrast, false alarm rate did not change with pupil size. The increase in hit rate can be explained by a participant either adopting a more liberal decision criterion or improving sensitivity, or both. To test these potential accounts, we calculated criterion and sensitivity (following the signal detection theory [SDT]; *Green and Swets, 1966*) within each group of trials sorted by prestimulus baseline pupil size. We observed that detection criterion linearly decreases (i.e., becoming more liberal) with increasing pupil size (*Figure 4C*). Sensitivity (*d'*) is related with prestimulus pupil size in an inverted U-shaped relationship, potentially following the classic Yerkes–Dodson law (*Table 1*). We also observed faster and more accurate object categorization responses following larger baseline pupil size (*Figure 4D*, *Table 1*). Data and model fit for individual subjects are plotted in *Figure 4—figure supplement 1*.

We next tested whether pupil-linked states influence perceptual behavior by shaping the neural representation of stimulus content. Such a prediction might be expected since larger baseline pupil size predicts higher categorization accuracy behaviorally (*Figure 4D*). To this end, we used a single-trial multivariate decoder (logistic regression) trained to predict stimulus category from MEG sensor-level stimulus-triggered activity. The MEG activity was averaged within consecutive 100 ms time windows and the decoders were fit for each time window and each pupil-linked group of trials separately, using a leave-one-out cross-validation scheme (*Figure 4E*). The effects of time from stimulus onset and pupil-linked baseline state on decoding accuracy were tested by a two-way repeated-measures ANOVA (angular transformation was applied to accuracy scores before entering into the ANOVA, see Materials and methods). As expected, neural representation of stimulus category significantly

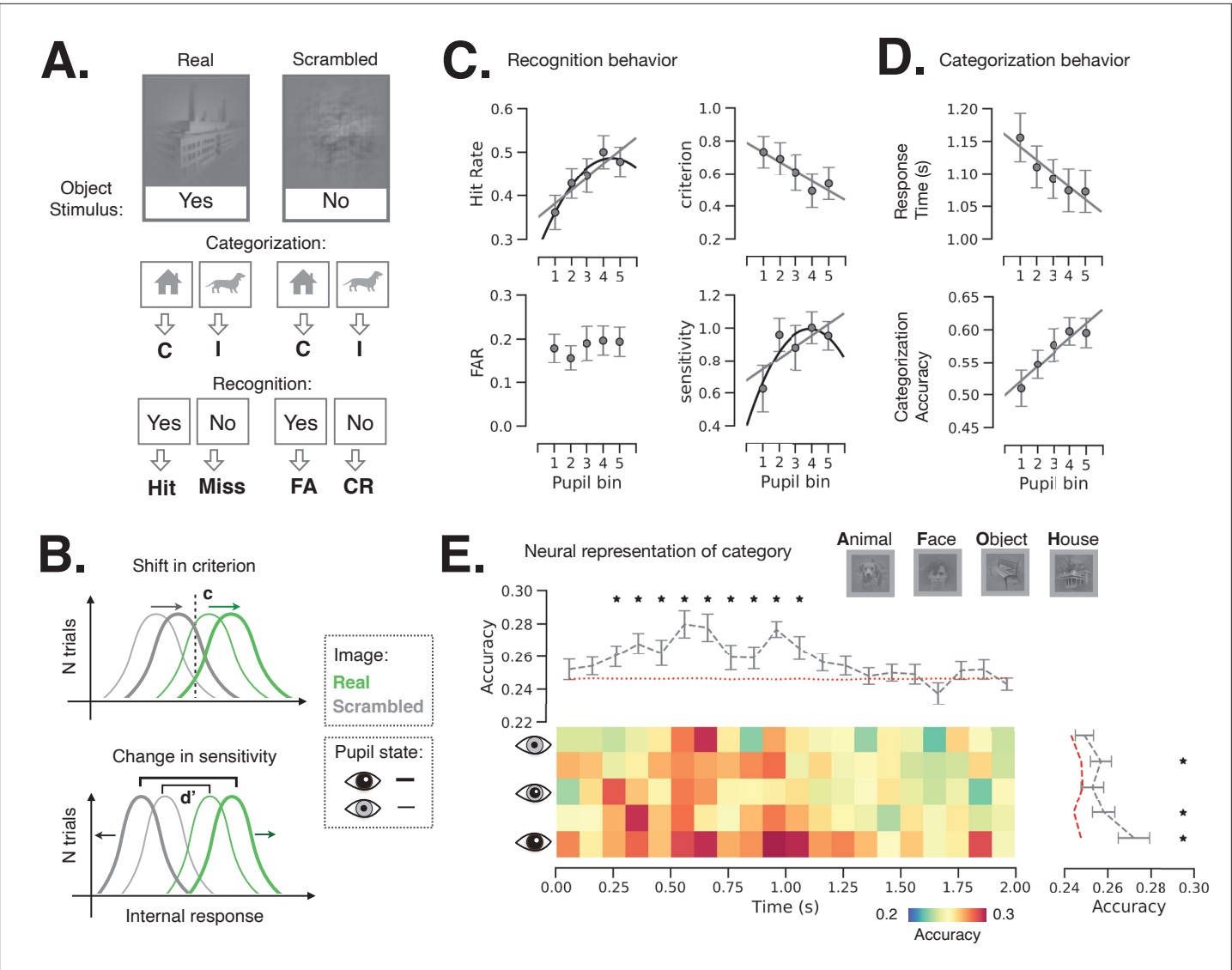

**Figure 4.** Behavioral and neural consequence of variation in baseline pupil-linked state. (**A**) Schematic of experimental stimulus types and their classification into behavioral metrics. C: correct, I: incorrect, FA: false alarm, CR: correct rejection. (**B**) Illustration of changes in detection criterion and sensitivity with prestimulus baseline pupil-linked state, summarizing results in (**C**). (**C**) Recognition behavior in a perceptual decision-making task in groups of trials sorted in bins according to prestimulus baseline pupil size. Statistical tests were performed using linear mixed model (LMM). Models with quadratic and linear or only linear terms were considered and plotted when the fit was significant (p < 0.05, full statistics are given in Table 1). (**D**) Categorization behavior: reaction time (top) and categorization accuracy (bottom) as a function of prestimulus baseline pupil size. (**E**) Single-trial decoding of stimulus category from whole-brain sensor-level stimulus-triggered magnetoencephalography (MEG) activity. The heatmap shows decoding accuracy in trials sorted according to prestimulus baseline pupil size (rows) and in 100 ms time windows following stimulus onset (columns). The top and right panels show averaged decoding accuracy across pupil-linked states or time; red dashed lines indicate the empirical chance level obtained through label permutations (K = 500) and asterisks indicate the time points/conditions where decoding accuracy was significantly better than chance (p < 0.05, FDR-corrected, label permutation test). Two-way repeated-measures ANOVA shows significant main effects of decoding time and prestimulus pupil diameter on decoding accuracy (p < 0.05) with no significant interaction. Error bars in all panels indicate s.e.m. across subjects.

The online version of this article includes the following source data and figure supplement(s) for figure 4:

**Source data 1.** Source data for *Figure 4C,D*, including individual subject-level data for perceptual behavior as a function of pupil size.

**Source data 2.** Source data for *Figure 4E*, including individual subject-level data for post-stimulus category decoding accuracy according to time point and prestimulus pupil size.

**Figure supplement 1.** Individual-subject data accompanying *Figure 4C,D,E*.

changed over time following the stimulus onset ($F$ = 3.81, p = 1.8 × 10⁻⁷). Importantly, pupil-linked states significantly affected the stimulus-triggered category representation ($F$ = 2.97, p = 0.03), with larger pupil size preceding better neural representation of stimulus category, consistently with a previous report (*Warren et al., 2016*). No effect of interaction between baseline pupil size and time from stimulus onset was observed ($F$ = 1.02, p = 0.43). In addition, the decoding accuracy was significantly above the chance level (obtained by label permutations) from 200 ms to 1 s after stimulus onset (p < 0.05, FDR-corrected) and significantly above the chance level for trials in the second, fourth, and fifth pupil size groups.

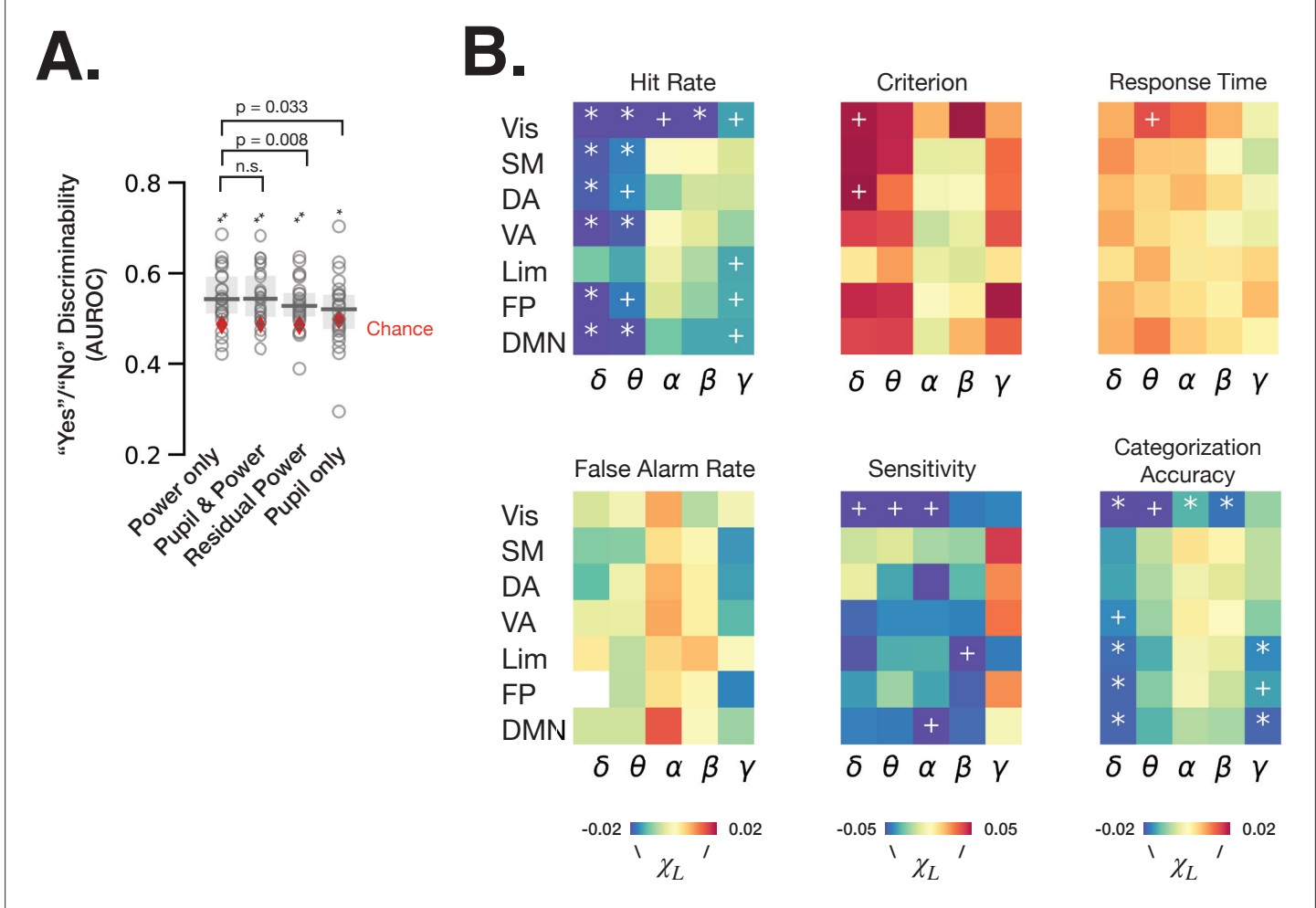

**Figure 5.** Spontaneous fluctuations of power shape behavior through both arousal-linked and arousal-independent mechanisms. (**A**) Performance of logistic regression models (quantified as AUROC [area under the receiver-operator curve]) predicting recognition behavior ('yes'/'no' reports) from prestimulus power in all frequency bands and resting state networks (RSNs) ('power only'), with prestimulus pupil size in addition to power ('pupil and power'), with residual power that was independent from pupil size ('residual power'), and with prestimulus pupil size as a single model feature ('pupil only'). Model performance was compared against chance through label permutation tests (* indicates p < 0.05, **p < 0.01 after FDR correction). 'Power only' model performance was compared with other models by Wilcoxon signed-rank test. (**B**) Parameter estimates of fitted linear mixed-effects model (LMM) where the trials were sorted according to the residual (independent-of-pupil) power. Each table element indicates parameter estimate for one frequency band and one RSN. * indicate p < 0.05 after FDR correction across RSNs, and + indicate p < 0.05 uncorrected (full statistics can be found in Table S2). Source data is available as a supplementary file.

The online version of this article includes the following source data for figure 5:

**Source data 1.** Source data for *Figure 5A*, including individual subject-level data for "Yes"/"No" discrimination.

**Source data 2.** Source data for *Figure 5B*, including individual subject-level data for residual power's relationship to perceptual behavior, separately for each RSN and frequency band.

## Arousal-linked and arousal-independent contributions of spectral power to behavior

To understand the respective contributions of spectral power that can be explained by pupil-linked arousal (*Figures 2–3*) and of spectral power independent of arousal to perceptual behavior, we first fit logistic regression models predicting 'yes'/'no' recognition behavior from baseline fluctuations of spectral power in all frequency bands and RSNs ('power only'), models with pupil size in addition ('pupil and power'), models using residual power independent of pupil size ('residual power') and models with pupil size as a single feature ('pupil only') (*Figure 5A*, see Materials and methods). The residual power was calculated by first fitting LMMs for each frequency band and each RSN for each participant, whereby the model with lower BIC at the group level was chosen between *equations 3* and *4* (see Materials and methods, *Table 1*). We find that all logistic regression models predict 'yes'/'no' behavior better than chance (permutation test, p < 0.05 FDR-corrected). Interestingly, the performance of 'power only' model does not improve by adding pupil size information (W = 143, p = 0.86, Wilcoxon signed-rank test), but the performance is worsened by regressing out the pupil-linked spectral power (W = 241, p = 0.008, Wilcoxon signed-rank test). These results indicate that both arousal-linked and arousal-independent spectral power fluctuations at baseline predict subsequent recognition behavior, but our recordings of pupil size did not provide additional information contributing to the prediction of behavior that was not already present in cortical spectral power.

Next, we aimed to clarify the effects of residual power within each frequency band and RSN on behavior. To this end, we sorted the trials into five groups according to the residual power in each frequency band and RSN, and calculated behavioral metrics for each group of trials. Using an LMM (Materials and methods, *equations 5–6*), we then determined whether pupil-independent spectral power in a given RSN could explain shifts in behavior. Analysis using the linear model (*Equation 5*) revealed pupil-independent negative correlation of power in delta and theta bands with hit rate, with the largest effect size in the visual RSN (*Figure 5B*, *Supplementary file 1*). Alpha and beta power in visual RSN also negatively correlated with hit rate. A negative correlation between alpha power and hit rate was previously reported in visual detection tasks and interpreted as spontaneous fluctuations of arousal (*Ergenoglu et al., 2004*). Our result, however, suggests that alpha power's effect on hit rate in our task is independent from pupil-linked arousal (in fact, alpha power's relation with pupil size predicts a positive correlation with hit rate, opposite to the experimental findings; see *Figures 3B and 4C*).

The effects on hit rate may stem from changes in criterion, sensitivity, or both. Using SDT analysis, we found that residual delta power in the dorsal attention network (DAN) and visual network has an effect on detection criterion, such that higher delta power results in a more conservative criterion. DAN includes the decision-making-related frontal eye field (FEF) and posterior parietal cortex areas (*Dorris and Glimcher, 2004*), which were shown to reflect decision criterion in non-human primates (*Gold and Shadlen, 2007*). Interestingly, detection sensitivity, but not criterion, was influenced by pupil-independent alpha power in the visual and default mode RSNs. This result complements the previous findings of visual alpha power influencing detection criterion instead of sensitivity (*Samaha et al., 2020*), and underscores the importance of considering pupil-linked state. In addition, visual delta and theta power and limbic beta power negatively correlate with sensitivity. Lastly, categorization accuracy is affected by pupil-independent prestimulus power several frequency bands and RSNs. In sum, spontaneous fluctuations in cortical spectral power contribute to perceptual decision-making independently of arousal-linked fluctuations.

Fitting the model type with quadratic and linear components (*Equation 6*) produced smaller BIC only in 9 out of 210 cases, indicated that the models with a linear component only is a better choice in 96 % of cases. Out of the nine quadratic models with the lower BIC, however, none produced a significant quadratic component (p > 0.05).

## Pupil size also reflects brain activity on a faster timescale, but with no discernable effect on behavior

Our analyses so far have revolved around slow (seconds) timescales of brain state fluctuations, in contrast to previous studies in animals showing that pupil size also changes with brain activity on a faster (milliseconds) timescale: specifically, pupil dilates after increased LC neuronal firing with an ~300 ms delay in non-human primates (*Joshi et al., 2016*). While signals from brainstem nuclei such

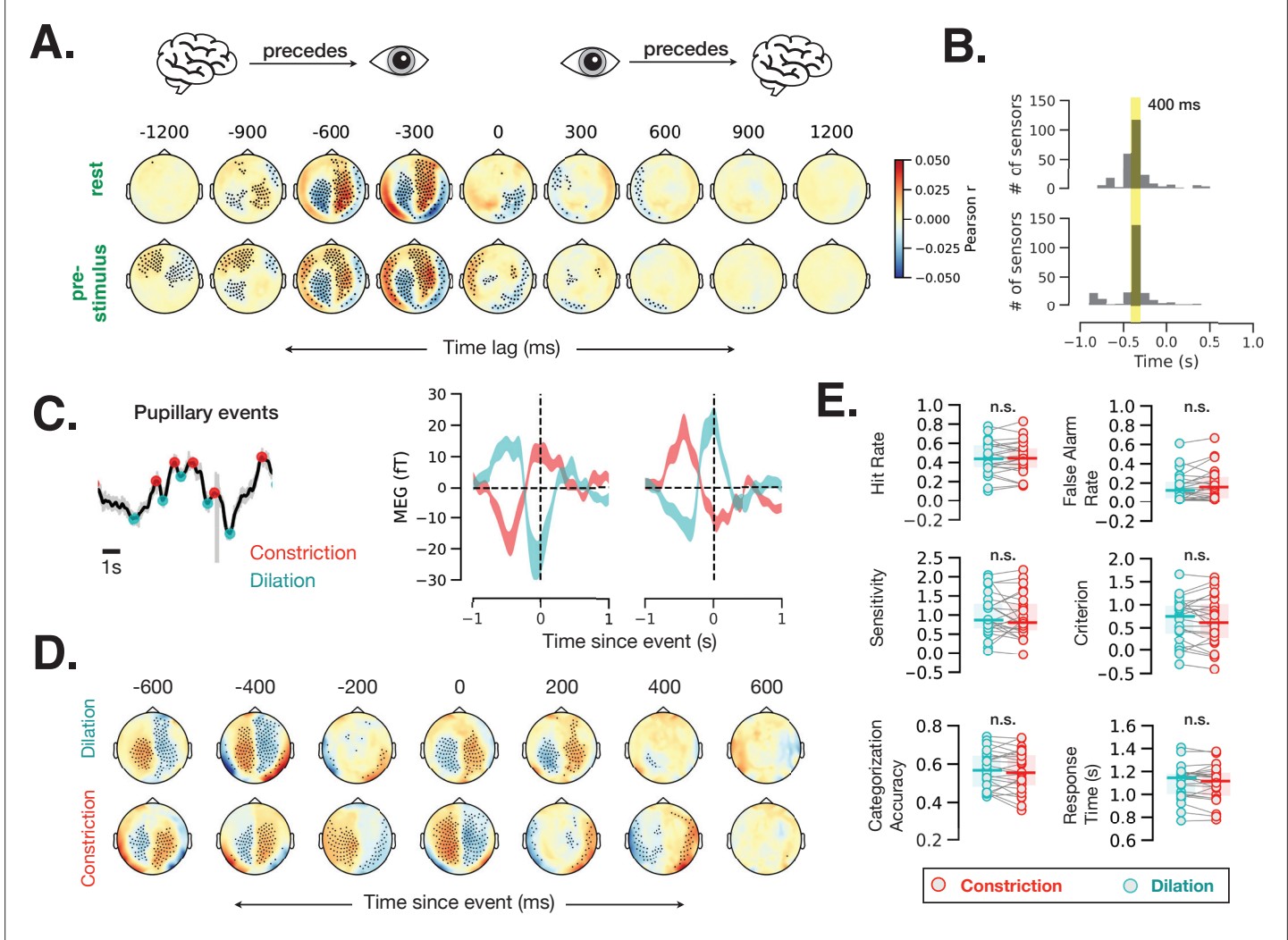

**Figure 6.** Fast (sub-second) timescale of pupil and brain interaction. (**A**) Cross-correlation between pupil size and sensor-level magnetoencephalography (MEG) activity during rest and prestimulus baseline. (**B**) Distribution of the cross-correlation peak and trough times in MEG sensors that showed significant correlation at any time point. Most sensors correlate with pupil size with a 400 ms delay. (**C**) Example of identified pupillary events, spontaneous constriction and dilation, and MEG activity triggered by such events in two occipital MEG sensors. (**D**) Same analysis as (**C**) across all sensors. Black dots in A and D indicate significant sensor-time locations (p < 0.05, spatiotemporal cluster-based permutation test). (**E**) A lack of behavioral consequence of stimulus presentation during a constriction or a dilation event. Constriction and dilatation are defined as a decreasing/ increasing pupil size in the 100 ms before stimulus presentation.

The online version of this article includes the following source data and figure supplement(s) for figure 6:

**Source data 1.** Source data for *Figure 6E*, including individual subject-level data for perceptual behavior sorted by prestimulus fast pupil dynamics.

**Figure supplement 1.** Additional analyses of dilation and constriction events.

as LC are difficult to detect with MEG, the brainstem nuclei send widespread cortical projections and cortical activity is readily detected by MEG. Thus, we hypothesized that covariation between pupil size and cortical activity at a fast timescale can be detected by MEG, which provides excellent temporal resolution (1200 Hz).

To estimate the time lag of correspondence between brain activity and pupil size, we computed time-resolved cross-correlation between each MEG sensor within each 2 s window during rest and prestimulus baseline with 100 ms time lags. Brain activity recorded from most MEG sensors correlated with pupil diameter at this fast timescale (*Figure 6A*; spatiotemporal cluster-based permutation test, p < 0.05). The correlation values were highest at a lag of 400 ms in most MEG sensors (*Figure 6B*), suggesting that cortical activity precedes fast pupil events by ~400 ms.

To investigate whether this prominent cross-correlation between pupil size fluctuation and MEG activity is related to the pupillary 'fast events'—spontaneous momentary constrictions and dilations (*Figure 1D*)—we calculated event-related field time-locked to fast pupil events in the resting state. *Figure 6C* shows example fast pupillary events (left) and averaged MEG activity time-locked to these constriction and dilation events in two occipital sensors (right). Both dilation- and constriction-related MEG activity shows one peak and one trough at ~0 and –400 ms. Control analyses confirmed that the distance between peaks, method of peak selection, or filtering the pupil time course do not influence the identified time courses of these fast pupil events (Materials and methods, *Figure 6—figure supplement 1*). *Figure 6D* shows MEG activity time-locked to fast pupillary events across the whole brain, with many sensors exhibiting significant activity preceding and following spontaneous pupil dilation and constriction. Assuming that human LC activity leads pupil changes with the same time delay as in monkeys, these findings are compatible with the possibility that cortical activity leads LC activity in the resting state (*Joshi et al., 2016*).

Together with previous animal findings (*Joshi et al., 2016*), our results in humans (*Figure 6A–D*) show that spontaneous pupil constrictions and dilations follow cortical and subcortical neural activity on a fast timescale (with a 300–400 ms lag). Do these fast, spontaneous pupil events influence perceptual behavior? To answer this question, we sorted the trials according to whether the stimulus was presented during a dilation or a constriction event (see Materials and methods, Eye-tracking and pupil size analysis) and calculated the behavioral metrics for these two groups of trials separately. We did not observe any difference in perceptual behavior according to whether the stimulus was presented during a dilation or a constriction event (*Figure 6E*). A control analysis using more stringent thresholds (50 % of trials with largest constriction or dilation slopes) yielded similar null results (p > 0.05 for all behavioral variables). Together, these findings suggest that only slower pupil-linked states shape perceptual decision-making, but not fast pupillary events, even though these fast events are associated with robust changes in neural activity.

## Discussion

In this study we characterized the relationship between spontaneous fluctuations in central and peripheral markers of arousal (cortical spectral power and pupil size) and delineated their contributions to behavior in a visual perceptual decision-making task. We uncovered that slow (~seconds) fluctuations in pupil size differentially relate to spontaneous large-scale changes in cortical spectral power across a broad range of frequencies. Spontaneous fluctuations of cortical activity power influenced perceptual behavior through arousal-linked and arousal-independent mechanisms. Lastly, transient pupillary events (constriction and dilation) were preceded by spontaneous cortical activity at a fast timescale (~400 ms) but did not have discernable influence on perceptual behavior.

### Spontaneous fluctuations in cortical spectral power are associated with pupil-linked arousal

Our data suggest that pupil-linked arousal covaries with spontaneous widespread changes in electrophysiological brain activity in humans (*Figures 2 and 3*, *Figure 3—figure supplements 1–4*, *Supplementary file 1*). This widespread arousal effect on cortical activity is consistent with previous human fMRI studies showing that large-scale fluctuations of BOLD signal in the resting state correlate with pupil size (*Yellin et al., 2015*; *Shine et al., 2016*; *Schneider et al., 2016*), as well as the widespread cortical projections from subcortical neuromodulatory systems including the LC-NE or BF-ACh systems. Activity of both systems correlates with pupil size and cortical activity in mice (*Reimer et al., 2016*) and non-human primates (*Aston-Jones and Cohen, 2005*, *Turchi et al., 2018*), but only limited data is available in humans (*de Gee et al., 2017*, *Murphy et al., 2014*). The full circuit-level mechanisms of these two systems are not fully understood and are currently under investigation (*McCormick et al., 2020*; *Joshi and Gold, 2020*). Additional subcortical neuromodulatory systems (such as dopaminergic and serotonergic) send widespread projections to the cortex (reviewed by *van den Brink et al., 2019*) and their relationship to spontaneous fluctuations in pupil size and cortical activity deserves further investigation.

Pupil-linked arousal has widespread effects in the frequency domain as well, influencing cortical activity power from delta to gamma bands. Such a cross-frequency effect is consistent with intracranial

EEG studies reporting that the spectral power of cortical activity follows a 1/*f* function, the exponent of which links the power across frequencies (*He et al., 2010*; *Podvalny et al., 2015*; *Freeman and Zhai, 2009*; *He, 2014*). Because the effect of pupil-linked arousal spans all frequencies, it likely stems at least partly from aperiodic brain activity, where the 1/*f* exponent decreases during high arousal sates, leading to simultaneous increase of high-frequency power and decrease of low-frequency power. Indeed, a recent study demonstrated that 1/*f* exponent of intracranial EEG power spectrum in the frequency range of 30–45 Hz is smaller (i.e., the power spectrum is shallower) in wakefulness than under anesthesia, pointing to the effect in the expected direction (*Lendner et al., 2020*). Furthermore, both BF-ACh (*Szerb, 1967*) and LC-NE (*Berridge and Waterhouse, 2003*) activity and increase in pupil size (*McGinley et al., 2015a*; *Reimer et al., 2014*; *Vinck et al., 2015*; *Stitt et al., 2018*) are associated with spectral modulation following the same principle: increase in high-frequency and decrease in low-frequency power. Importantly, our findings extend these previous findings from single cortical areas to large-scale cortical networks and from animal models to humans. In addition, we report a more complex and nonlinear relationship of spectral power and pupil size than was previously known.

We observed an inverted U-shaped relationship between alpha power and pupil size in the resting state and prestimulus baseline, consistent with recent EEG studies (*van Kempen et al., 2019*; *Hong et al., 2014*). These previous studies only investigated the alpha band, while our results show that pupil-linked arousal correlates with global electrophysiological power shift between low- and high-frequency bands, whereby the alpha band appears to constitute the intermediate transition point. This relationship between prestimulus alpha power and pupil size is consistent with a previous suggestion that alpha power positively correlates with tonic alertness (*Sadaghiani and Kleinschmidt, 2016*; *Sadaghiani et al., 2010*), which might correspond to the dominant left-hand side of the inverted-U function (*Figure 3B*); further, the inverted U-shaped relationship we observed is consistent with the disappearance of alpha oscillations during drowsiness and light sleep—states with very low arousal (*Cantero et al., 1999*).

The brain activity modulation by pupil-linked arousal systems, while similar across RSNs, is not entirely uniform. We observed, for example, differences in parameter estimates across networks and behavioral conditions: modulation of alpha power by arousal in the visual RSN is stronger than in other networks at rest, whereas modulation of beta power is strongest in the DAN in task baseline (*Figure 3C*, linear components). Such non-uniform effects are consistent with the intriguing proposals of pupil-linked arousal systems playing an important role in dynamic reconfiguration of large-scale networks topology responding to cognitive demands (*van den Brink et al., 2019*; *Shine, 2019*). Indeed, while the arousal neuromodulatory systems send widespread projections to cortex, recent studies also report complex topographic organization which may produce such non-uniform modulatory effects (*Uematsu et al., 2017*; *Chandler et al., 2019*). A greater understanding of the function of uniform vs. modular effects on cortical power by arousal systems in various cognitive states is an important goal for future investigation.

## Behavioral relevance of pupil-linked brain state

We found that prestimulus pupil-linked brain state influences both the detection sensitivity (*d′*) and criterion (*c*) in a visual perceptual decision-making task (*Figure 4*). While the relationship between pupil size and behavior received much attention in the last century, most of the early studies dismissed the importance of prestimulus pupil size's effect on subsequent behavior (*Goldwater, 1972*). The effect of prestimulus pupil size on perceptual behavior was evaluated in several recent studies. A previous study (*Waschke et al., 2019*) using an auditory discrimination task observed a similar inverted-U function between perceptual sensitivity and prestimulus pupil size as we observed here. A null effect of baseline pupil size on behavior was reported in a previous visual detection study (*de Gee et al., 2017*), but the analyzed baseline period in this study coincided with the appearance of a noise stimulus, which may have influenced pupil diameter during the baseline period. Lastly, we found an opposite effect of prestimulus pupil size on reaction time as compared to an earlier study (*van Kempen et al., 2019*); this difference might result from different task designs and/or the two studies sampling different regions of arousal variation that constitute the left vs. right arm of a U-shaped function.

Recent animal studies report effects of baseline pupil size on behavior as well. An inverted U-shaped relationship between baseline pupil size and subsequent hit rate and detection sensitivity was reported in the mouse (*McGinley et al., 2015a*; *Steinmetz et al., 2019*), consistent with our findings. Criterion was reported to follow a U-shaped function with baseline pupil size (*McGinley et al., 2015a*); our data is consistent with the left-hand side of this relationship, potentially due to the mouse achieving higher arousal levels during the experiment. Future human studies involving higher arousal states will allow a more comprehensive comparison between mouse and human studies.

Extending the classic Yerkes–Dodson law, the adaptive gain theory (*Aston-Jones and Cohen, 2005*) postulates that ongoing changes in pupil size reflect transitions between several states: exploration, when the tonic LC activity is high and the pupil is large; exploitation, when the LC activity is bursty and the pupil is of intermediate size; and drowsiness, when the tonic LC activity is low and the pupil is small. Here, we observed the lowest perceptual sensitivity with the smallest pupil size (*Figure 4C*), which might be explained by low arousal or drowsiness. Sensitivity was higher with intermediate pupil size, potentially due to transition to the state of exploitation. The largest pupil recorded in our study corresponded to the decay phase of the quadratic model, potentially indicating a decreasing sensitivity with the transition to the exploratory behavioral state. Interestingly, the largest prestimulus pupil also cooccurs with a subsequent shift to a more liberal decision criterion, which intuitively fits with the adaptive gain theory framework because exploration requires accepting a less rewarding option. It has been shown indeed that a larger baseline pupil diameter precedes more exploratory choices (*Jepma and Nieuwenhuis, 2011*). Further research on a link between the exploration-exploitation behavior according to the adaptive gain theory and detection criterion and sensitivity according to SDT could constitute an exciting new direction as it seems that both types of behavioral state transitions could be controlled by the same underlying mechanism.

## Behavioral consequences of spectral power associated with and dissociated from pupil-linked arousal

Interestingly, recognition behavior ('yes'/'no' reports) could be predicted better than chance from single-trial baseline spectral power and, while pupil size did not provide additional predictive power to the model, removing the pupil-linked spectral power fluctuations significantly reduced the model performance (*Figure 5A*). This result indicates that behavior in a visual perceptual decision-making task is influenced by both arousal-linked and arousal-independent fluctuations of cortical spectral power.

After controlling for the effect of pupil-linked arousal, we found that alpha power in the visual network predicted detection sensitivity but not criterion (*Figure 5B*). A large body of previous work has shown a link between prestimulus alpha power and perceptual behavior, (for example, *van Dijk et al., 2008*; *Limbach and Corballis, 2016*; *Iemi et al., 2017*), and the predominant pattern from these studies is that spontaneous fluctuations of alpha power in visual areas influence detection criterion (and associated subjective visibility) but not sensitivity (reviewed recently in *Samaha et al., 2020*). These previous studies, however, did not account for behavioral modulation by pupil-linked arousal states, which, as we have shown, influence most frequency bands. Our results indicate that, first, prestimulus alpha power is modulated by both pupil-linked and pupil-independent mechanisms; and second, the pupil-independent alpha power fluctuation appears to affect perceptual sensitivity but not criterion, while pupil-linked arousal influences both sensitivity and criterion.

Previous studies have also shown that alpha power modulation plays an important role in attentional gating, such that alpha power suppression corresponds to the allocation of spatial- and feature-based attention (*Foxe and Snyder, 2011*; *Jensen and Mazaheri, 2010*). Given that attentional cueing typically increases detection accuracy (*Haegens et al., 2011*; *Voytek and Knight, 2010*) and sensitivity ($d'$) (*Kelly et al., 2009*; *Smith, 2000*), our observed negative relationship between pupil-independent alpha power and detection sensitivity is consistent with the attention effect. However, because our task did not explicitly manipulate spatial- or feature-based attention, how the pupil-independent alpha power fluctuations relate to attention-related alpha modulations, and whether their influences on perceptual behavior share the same underlying mechanism, remains to be tested.

After controlling for the effect of pupil-linked arousal, we also found that prestimulus delta power (1–4 Hz) in the DAN and visual network influences subsequent detection criterion, but not sensitivity, with higher delta power preceding a more conservative criterion. This finding is inconsistent with

previous studies reporting a null effect of prestimulus EEG delta power on visual (*Ergenoglu et al., 2004*) or auditory (*Waschke et al., 2019*) stimulus detection. Consistent with our result, spontaneous fMRI activity in DAN was shown to correlate with subsequent stimulus detection in humans (*Boly et al., 2007*; *Sadaghiani et al., 2009*). Previous fMRI research also revealed an effect of prestimulus DAN activity on subjective perceptual bias in a random dot motion detection task (*Rahnev et al., 2012*). DAN includes the decision-making-related FEF and posterior parietal cortex (*Dorris and Glimcher, 2004*), which were shown to reflect decision criterion in non-human primates (*Gold and Shadlen, 2007*); electrical stimulation of FEF also affects the pupil size, potentially due to mechanisms related to selective attention (*Ebitz et al., 2017*). Together, the effect of delta power in DAN on detection criterion is consistent with previous human fMRI and non-human primate electrophysiology studies; but our study specifically rules out pupil-linked arousal in this effect and provides a frequency domain localization to the delta band.

## The timescale of pupil size fluctuations and momentary events

Pupil size correlates with large-scale MEG activity on a faster timescale as well: first, spontaneous variations in MEG activity precede spontaneous variations in the pupil size with a lag of 400 ms (*Figure 6A–B*); second, pupillary events (constriction and dilation) coincide with MEG activity peak at lag zero and are preceded by an MEG activity peak occurring 400 ms earlier (*Figure 6C–D*). A similar pattern of two peaks (at 0 and –300 ms) was observed in monkey LC spike rates (*Joshi et al., 2016*). The zero-lag peak we observed could result from a neural event occurring earlier in time and influencing both pupil size and MEG activity at the same time lag. Since MEG activity originates mostly in the cortex, the peak at –400 ms lag (100 ms earlier than the peak in monkey LC) could point to an influence of cortical resting state fluctuations on LC activity which then triggers pupillary events. Such a potential influence, however, does not seem to be behaviorally relevant: whether a liminal visual stimulus is presented during ongoing pupil dilation or constriction did not influence perceptual behavior in our task (*Figure 6E*). This result is surprising considering that pupil dilation and constriction events are controlled by sympathetic and parasympathetic pathways, respectively, activation of which is considered to have distinct effects on behavior ('fight or flight' vs. 'rest and digest'). On the other hand, the sympathetic and parasympathetic pathways are also indirectly affected by the shared neuromodulatory central arousal systems (*Lowenstein et al., 1963*), and it is possible that in our task the pupillary dilation and constriction events merely reflect the adjustment of this central arousal state, not reaching the threshold of unique sympathetic or parasympathetic effects on behavior. A previous study in the mouse reported that orientation tuning in V1 is better when a stimulus is presented during pupillary dilation as compared to constriction (*Reimer et al., 2014*); however, it remains unknown if such improved tuning affects behavior. Future research is needed to fully map out the relationship between pupil-linked state and behavioral measurements across different timescales and in variety of tasks.

## Limitations and future directions

First, the continuum of behavioral states studied herein is inherently limited. Animals and humans in natural settings exhibit states of much lower arousal, such as deep sleep, and of much higher arousal, such as escape from danger. Understanding how such extreme states extend the link between pupil size and spectral power will be crucial to understanding the full range of state modulation of perceptual behavior. Second, spontaneous fluctuations in pupil size on a timescale longer than a run length (5.6 ± 0.38 min) might be important. In the present experiment such longer timescales could not be assessed since we used a relative measure of pupil size change within each experiment block, whereas an absolute pupil size measure (not available using the present eye-tracking set-up) is necessary to assess change in pupil size across blocks. Third, while the quadratic model we used herein provides an excellent fit to our data, it offers a simplified view of human brain state regulation. Future work focusing on normative approaches to understanding state regulation (e.g., considering optimal states for behavior) may lead to alternatives by expanding the model families and their parameters. For example, models from sigmoidal or Gaussian families might be more biologically plausible. Finally, the full composition of endogenous events driving spontaneous fluctuations in pupil size remains incompletely understood. While a causal association between activity in brainstem neuromodulatory centers and pupil size has been established, additional endogenous factors, ranging from cognitive

(e.g., an exciting thought) to physiological (e.g., blinks), affect the pupil size. For example, it has been shown that blinks correlate with a subsequent fluctuation in pupil size lasting for up to 4 s (*Knapen et al., 2016*). Importantly, it is unclear if such endogenous effect can be dissociated from arousal or not. Given such a long-lasting effect, excluding trials with blinks within 4 s intervals is not feasible (normal human blink rate can reach ~1/3 s). Thus, future research is needed to determine to what extent different endogenous events drive the spontaneous fluctuations in pupil size in addition to the known neuromodulatory effects.

## Implications

Our first aim in this study was to fill in the missing link between two well-known markers of human arousal—pupil size and cortical spectral power. The models characterizing this relationship will serve the research community in need to interpret pure eye-tracking studies in terms of neurophysiological mechanisms shaping behavior. In addition, our findings provide insights enabling the interpretation of human findings in the context of rapidly advancing knowledge about circuit-level mechanisms of arousal in animal models. Our second aim was to delineate the contributions of prestimulus pupil-linked arousal and cortical spectral power to human perceptual behavior. Our findings shed new light on the functional significance of prestimulus brain state at large, showing that both pupil-linked and pupil-independent spontaneous activity serves more than one function in human perceptual behavior. By presenting the distinct behavioral consequences of different prestimulus brain states, we begin to uncover the rich variety of spontaneous neural processes shaping behavior.

## Materials and methods

### MEG data acquisition

MEG data were recorded at a sampling rate of 1200 Hz using a 275-channel scanner (CTF, VSM MedTech). Before and after each block, the head position of the subject was measured using coils placed on the ear canals and the bridge of the nose. Between blocks, the head position of the subject was measured with respect to the MEG sensor array using coils placed on the left and right preauricular points and the nasion, and the subject self-corrected their head position to the same position recorded at the start of the first block using a custom visual-feedback program in order to minimize head displacement across the experiment. The task data analyzed herein have been used in a previously published report (*Podvalny et al., 2019*), which included multivariate models of prestimulus sensor-level MEG activity predicting forthcoming perceptual decisions according to content-specific and non-content-specific ongoing processes. The data from rest periods analyzed in this paper have not been previously published, and the questions addressed in this study are distinct from previous publication.

### MEG data preprocessing

Three dysfunctional sensors were removed from all analyses. Independent component analysis was performed on each experiment run to remove blink, cardiac, and movement-related artifacts. The linear trend was removed from each experiment run. No frequency domain filtering was applied in order to avoid artifactual signal bleeding from the post-stimulus signal into the prestimulus period. MEG data were preprocessed using Python and the MNE toolbox (*Gramfort et al., 2014*) (version 0.19.1).

### Participants

All participants (*N* = 25, 15 females, mean age 26, range 22–34) provided written informed consent. The experiment was approved by the Institutional Review Board of the National Institute of Neurological Disorders and Stroke (protocol #14 N-0002). The participants were right-handed, neurologically healthy, and had normal or corrected-to-normal vision. One enrolled participant decided to stop the experiment after finishing one experiment block due to discomfort and is not included in data analyses. Three subjects did not perform the second 5 min rest recordings due to time constraints.

## Rest

Participants were instructed to fixate at a crosshair in the middle of a gray screen and to avoid meditating or engaging in repetitive mental activity, such as counting. We recorded one 5 min session before the task onset in all 24 participants and one 5 min session after the task in 21 participants (second session recording of three participants was omitted due to time constraints).

## Task

Participants engaged in a threshold object detection task where they were instructed to report object category (face, house, object, animal) and their subjective recognition experience ('yes' or 'no') following a briefly presented low-contrast object stimulus. Subjective recognition experience has been defined as seeing an actual object versus seeing a noise pattern or seeing nothing at all. Prior to the beginning of the main task, we used an adaptive staircase procedure 'QUEST' (*Watson and Pelli, 1983*) to identify image contrast resulting in subjective recognition rate of ~50 % 'yes' reports for the same image. The main task included 300 real object images and 60 images of their phase-scrambled counterparts and was conducted in 10 experiment runs with self-paced breaks. The subjects were instructed to fixate on the center crosshair during the prestimulus interval and to avoid blinking to the best they can until after stimulus arrival. Each unique image was presented at a staircase-determined contrast and was identical across trials. Task details are also fully available in our previous publication (*Podvalny et al., 2019*).

## Data epoch analysis

Continuous data (rest) were cut into 2 s segments (epochs), non-overlapping in time, such that 5 min recording led to 149 epochs. Task data was cut in 2 s fragments before each stimulus onset, which led to 360 epochs, corresponding to number of trials.

## Eye-tracking and pupil size analysis

The recordings were made in a dimly lit room with a screen as an additional source of luminance. Subjects' pupil size was continuously monitored using a video-based eye-tracking system (EyeLink 1000+), in the binocular mode with a sampling rate of 1000 Hz. Only right eye recordings were used in the analysis since no difference in pupil size between eyes was expected in healthy volunteers. Blinks or missing data periods were detected by identifying the time points where the recorded pupil size of the right eye dropped below a constant threshold. The missing data detection threshold was determined for each participant by visual inspection of data and was between –3.5 and –4 in EyeLink provided arbitrary units. Blink onset was defined as 100 ms before crossing the threshold and blink offset was defined as 100 ms after. 'Slow states' were calculated as a simple averaged pupil size in consecutive 2 s interval in rest and 2 s before stimulus onset in task, with the blink periods excluded from average calculation. The averaged prestimulus pupil size then was z-scored within each block. 'Fast events' were calculated on pupil time course after blink interval interpolation using piecewise cubic Hermite interpolating polynomial. Next, in order to remove potential high-frequency artifacts, we applied a 5 Hz low-pass Butterworth filter to continued resting state data forward and backward in time using *filtfilt* function (*Scipy*), assuring no phase shifts. Peaks and troughs were defined as time points where constriction and dilation begin respectively. To identify peaks/troughs, for each data point in the pupil size time course, we tested whether its amplitude is the highest/lowest in a 1 s time window centered on that data point (using 'argrelextrema' method of *scipy*). Finally, to calculate whether a stimulus is presented during ongoing constriction or dilation, we fit a linear regression to pupil size in a 100 ms window before stimulus onset. To make sure the linear fit is less susceptible to high-frequency artifacts, a 5 Hz low-pass filter was applied on prestimulus data only (using *MNE epochs.filter* zero-phase FIR with Hamming window). A negative slope of the line indicates constriction and a positive slope indicates dilation.

## Forward model

We constructed the forward model using structural MRI scans (1 mm isotropic voxels, MP-RAGE sequence), acquired on a either 3T Siemens Skyra (Siemens, Erlangen, Germany), General Electric 3T scanner with an 8-channel head coil or Siemens 7T MRI system equipped with a 32-channel head coil (Nova Medical, Wilmington, MA). Images were processed in Freesurfer (recon-all). Skull strip and pial

surfaces were inspected and manually corrected if necessary. We used a boundary element method (Watershed method) to create a head shape and then aligned it with MEG coordinate system according to fiducial markers (nasion, left and right preauricular points, using MNE coreg tool). For two subjects no MRI scans were available and for two additional subjects the MRI quality was too low to identify the brain surfaces. For these four subjects, we used another subject's MRI as a template. A realistic single-shell brain volume conduction model was then constructed for each participant, based on these structural MRIs.

## Spectral analysis in source space

To localize power changes of pre-defined frequency bands, we used DICS technique (*Gross et al., 2001*) implemented in MNE-Python, following the analysis pipeline of *van Vliet et al., 2018*. DICS is a linearly constrained minimum variance beamforming method applied to time-frequency transformation of the original signals. Cross-spectral density (CSD) matrix was computed between signals recorded with each pair of MEG sensors using a multitaper method in frequency range of 1–100 Hz (DPSS taper windows). CSD was computed for each 2 s epoch defined in resting state or prestimulus period during task recordings. Next, a beamformer spatial filter is created for each frequency band using CSD matrix averaged across all epochs with a regularization parameter of 0.05. The frequency bands were defined as follows: delta 1–4 Hz, theta 4–8 Hz, alpha 8–13 Hz, beta 13–30 Hz, gamma 30–90 Hz. These filters were applied to the CSD matrices that were averaged across epochs corresponding to each pupil-linked group and each frequency range to obtain the group source power maps. The source power maps were then transformed to common space and submitted to group analysis.

## Analysis of behavior

We employed SDT to quantify task behavior. We first calculated hit rate (HR) and false alarms rate (FAR) as fraction of real-image trials and scrambled-image trials that were reported as recognized (i.e., the response to the question about object recognition was 'yes'), respectively. We implemented Macmillan and Kaplan correction (*Macmillan and Kaplan, 1985*) of FAR = 0 and HR = 1: the FAR was corrected to $\frac{1}{2N_{scr}}$ in the case of no FA trials, where $N_{scr}$ is the total number of scrambled-image trials; the HR was corrected to $1 - \frac{1}{2N_{real}}$ in the case of HR equal to 1, where $N_{real}$ is the total number of real-image trials. Next, we calculated measures of sensitivity (*d'*) and bias (*c*) following standard SDT analysis (*Green and Swets, 1966*) using subjective reports of recognition. *d'* indicates the ability to discriminate between real images containing objects and scrambled images that do not contain objects but preserve low-level features of the object images. It is computed by subtracting the Z-transformed FAR from the Z-transformed HA:

$$d' = Z\left(HR\right) - Z\left(FAR\right) \tag{1}$$

where *Z* is an inverse normal cumulative distribution function. *c* criterion represents the tendency to make 'yes' reports to indicate recognition, regardless of whether the stimulus is a real or scrambled image and is computed as follows:

$$c = -\tfrac{1}{2}\left(Z\left(HR\right) + Z\left(FAR\right)\right) \tag{2}$$

Categorization accuracy was simply calculated as a fraction of trials with correctly reported category. The time from the first question appearing on the screen to the button press corresponding to the categorization response was measured as the reaction time.

## Statistical modeling

We used LMMs with the maximal random effects structure justified by the design (*Barr et al., 2013*). LMMs allow modeling of data with repeated measures, where we considered the participants as a random effect on the intercept. All the non-categorial predictors were z-scored before fitting the models. The p-values of the model parameters were calculated via Wald tests and were corrected across RSNs using Benjamini–Hochberg procedure (*Benjamini and Hochberg, 1995*) controlling for false discovery rate. The models were fit using Powell's algorithm (*Powell, 1964*) minimizing standard

(rather than restricted) likelihood to assure the meaningful information criterion calculation. BIC was used for comparison between linear and quadratic models. The models were implemented using *statsmodels* (**Seabold and Perktold, 2010**) Python toolbox. We verified that the parameters estimates acquired with statsmodels were nearly identical to the ones acquired with a more widely used R lme4 (**Bates et al., 2015**) toolbox implemented through Python wrapper pymer (**Jolly, 2018**). We used piecewiseSEM (**Lefcheck and Freckleton, 2015**) R package to estimate marginal and conditional variance-explained values indicating the proportion of total variance explained by fixed effects only and the proportion of variance explained by both fixed and random effects, respectively (**Nakagawa et al., 2017**). It is important to note, however, these metrics were developed specifically for LMMs and are useful in the relative sense for comparison between such models' fit to the data but they do not have the same properties as linear models $R^2$.

*First,* to model the relationship between pupil size and spectral power in rest and prestimulus task intervals we used both a model with a linear and a quadratic term and a model with a linear term only, defined as follows:

$$Power_{i,j} \sim \chi_Q \cdot Pupil_{i,j}^2 + \chi_L \cdot Pupil_{i,j} + \chi_o + \gamma_{Qi} \cdot Pupil_{i,j}^2 + \gamma_{Li} \cdot Pupil_{i,j} + \varepsilon_{i,j} \tag{3}$$

$$Power_{i,j} \sim \chi_L \cdot Pupil_{i,j} + \chi_o + \gamma_{Li} \cdot Pupil_{i,j} + \varepsilon_{i,j} \tag{4}$$

where $\chi_Q$ and $\chi_L$ are fixed effects parameters (shared by all subjects) and $\gamma_{Qi}$ and $\gamma_{Li}$ are random effects parameters for subject *i*, and *j* represents the individual measurement in a 2 s interval. The model parameters with lower BIC are reported in all figures and **Supplementary file 1**. Spectral power was log-transformed to bring its distribution close to normal and the mean was removed for each subject (hence no random intercepts in the model: $\gamma_{oi} = 0$). Pupil size was z-scored to produce standardized effect size.

*Second,* we used an LMM to assess whether behavioral metrics change with pupil or with residual power. To this end, we first split the experimental trials into five groups according to quintiles of the distribution of pupil size or residual power (i.e., the groups contained equal number of trials (20%), see **Figure 1G**). The split was necessary since the calculation of the SDT metrics of behavior requires a group of trials. Within each group we calculated six behavioral variables: HR, FAR, *d'*, *c*, accuracy, reaction time (see the definitions above). We used two types of models, one with linear component only and one with both linear and quadratic components to study how behavior depends on the trial grouping:

$$Behavior_{i,G} \sim \chi_L \cdot G + \chi_o + \gamma_{Li} \cdot G + \gamma_{oi} + \varepsilon_{i,G} \tag{5}$$

$$Behavior_{i,G} \sim \chi_Q \cdot G^2 + \chi_L \cdot G + \chi_o + \gamma_{Qi} \cdot G^2 + \gamma_{Li} \cdot G + \gamma_{oi} + \varepsilon_{i,G} \tag{6}$$

here $\chi_Q$ and $\chi_L$ are fixed effects parameters (shared by all subjects) and $\gamma_{Qi}$ and $\gamma_{Li}$ are random effects parameters for subject *i*, and *G* represents the group of trials (1–5) the behavioral metrics were calculated for. The group numbers were z-scored to produce standardized effect size.

*Finally,* to calculate residual prestimulus spectral power in various frequency bands and RSNs, we fit a model with predictors given in **equations 3** or **4**, selected based on lower BIC, individually for each subject and submit its residual power to analysis of behavior using the models defined in **equations 5** and **6**, after calculating the behavior in trial groups split according to this residual power.

## Decoding

Logistic regression models were used to predict the presented object category or the recognition ('yes'/'no') reports on the single-trial level using a leave-one-out cross-validation scheme. The model was fit to all trials except one that was used for determining the accuracy of model's prediction and this leave-one-out cross-validation procedure was repeated for each trial in a group. To decode *object category*, the models were fit to averaged MEG sensor-space signal in each consecutive 50 ms window in the period of 0–2 s from stimulus onset, with trials split into five groups according to prestimulus baseline pupil size quintiles (0–20%, 20–40%, 40–60%, 60–80%, 80–100%). The effects of pupil-linked group and of time window from stimulus onset on model's prediction accuracy were tested using a repeated-measures ANOVA. The proportion of correct predictions made by the model

was transformed from binomial to normal distribution using angular transformation before entering into the ANOVA: $arcsine\left(\sqrt{proportion}\right)$ . To determine the empirical chance level, we shuffled the category labels and re-computed the decoding accuracy 500 times for each logistic regression model. We then calculated the proportion of times the accuracy predicted from actual data exceeded the permutation-derived chance level to assess the decoding significance. To decode the 'yes'/'no' report we used four types of models: (a) 'power only' included the estimates of power in all frequency bands and RSNs at baseline as model's features; (b) 'pupil and power' model included the same features as (a) and an additional feature of pupil size; (c) 'residual power' model included the residual power in all frequency bands and RSNs unexplained by pupil diameter fluctuations; (d) 'pupil only' model included the pupil size as a single feature. Model performance was quantified by area under the ROC (receiver-operator curve) for each subject and compared using Wilcoxon signed-rank tests.

## Pupil-MEG lagged correlation

For each 2 s epoch, a Pearson correlation coefficient was computed with 100 ms time lags between pupil diameter and activity at each MEG sensor. The correlation coefficients were Fisher-transformed and averaged across epochs for each participant and these averages were submitted to second-level group analysis. To assess significance of correlation coefficients at a group level and to address multiple comparisons, we employed a non-parametric spatiotemporal cluster-based permutation test (*Maris and Oostenveld, 2007*) using 1000 permutations and sensor-level and cluster-level thresholds of 0.05 (MNE toolbox implementation).

## Acknowledgements

We thank Navin Kariyawasam for help with literature review. This research was supported by an NSF CAREER Award (grant ID: BCS- 1753218; to BJH).

## Additional information

### Funding

| Funder | Grant reference number | Author |
| --- | --- | --- |
| National Science Foundation | BCS- 1753218 | Biyu J He |

The funders had no role in study design, data collection and interpretation, or the decision to submit the work for publication.

### Author contributions
Ella Podvalny, Conceptualization, Formal analysis, Investigation, Software, Visualization, Writing - original draft, Writing - review and editing; Leana E King, Formal analysis, Investigation, Visualization; Biyu J He, Conceptualization, Funding acquisition, Project administration, Resources, Supervision, Writing - original draft, Writing - review and editing

### Author ORCIDs
Ella Podvalny (iD) http://orcid.org/0000-0002-6810-2770
Biyu J He (iD) http://orcid.org/0000-0003-1549-1351

### Ethics
Human subjects: All participants provided written informed consent. The experiment was approved by the Institutional Review Board of the National Institute of Neurological Disorders and Stroke (protocol #14-N-0002).

### Decision letter and Author response
Decision letter https://doi.org/10.7554/eLife.68265.sa1
Author response https://doi.org/10.7554/eLife.68265.sa2

## Additional files

**Supplementary files**
• Supplementary file 1. Supplementary tables 1 and 2 containing detailed statistics related to *Figures 3 and 5*.

• Transparent reporting form

**Data availability**
Source data are available as csv files for all figures except for whole brain images. Analysis code supporting this study is available at a dedicated Github repository https://github.com/BiyuHeLab/eLife_Podvalny2021, copy archived at https://archive.softwareheritage.org/swh:1:rev:0422a1992d1dfcfd14bfa4403dac7f50668c831c.

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
