## [Decision Letter]

**Acceptance summary:**

This study takes a comprehensive look at the relationship between pupil-indexed arousal and cortical MEG activity and performance on a demanding sensory behaviour in humans subjects. Understanding how pupil activity, widely interpreted as a proxy of arousal, ties in with the dynamics of neural activity and behaviour is a key challenge in systems and cognitive neuroscience. The authors identified a widespread component of cortical activity that can be explained by changes in pupil dilation and which in turn predicts performance in participants' decision-making. This work will help build an important bridge between animal models of arousal and human cognition and behaviour.

**Decision letter after peer review:**

[Editors’ note: the authors submitted for reconsideration following the decision after peer review. What follows is the decision letter after the first round of review.]

Thank you for submitting your work entitled "Spectral signature and behavioral consequence of spontaneous shifts of pupil-linked arousal in human" for consideration by *eLife*. Your article has been reviewed by 3 peer reviewers, and the evaluation has been overseen by a Reviewing Editor and a Senior Editor. The following individual involved in review of your submission has agreed to reveal their identity: Jan Willem de Gee (Reviewer #3).

Our decision has been reached after consultation between the reviewers.

The reviewers and editors felt that this manuscript aims at offering important insights. First, brain-wide relationships across a wide range of spectra have only rarely been tested in humans. Second, testing direct and indirect links between baseline pupil size, spectral power and behaviour, separately, has not been done before in humans, to the best of our knowledge. Finally, the authors try to combine and compare the behavioural relevance of slow as well as fast pupil size fluctuations, linking to previous research in non-human animals.

Our enthusiasm was tempered, however, as the methodical issues raised by all reviewers are substantial. The editors were not convinced that these are outweighed by the potential advance in the results. The main concerns tie into issues raised by all reviewers about what appear to us as somewhat arbitrary decisions in how the models were formulated. The individual reviews, attached below, will delineate these concerns in greater detail.

*Reviewer #3:*

I think this is an important paper. The characterization of the relationship between baseline pupil size and large-scale cortical spectral power across a wide range of frequencies (localized using Dynamic Imaging of Coherent Sources) is state of the art and novel. However, I do have a number of serious issues with the methods. I expect at least some of the results will change when the authors address these.

I have a number of issues about the mixed effects modeling.

– The authors write that they only considered random intercepts, and not random slopes. Why? See for example Barr et al., Journal of Memory and Language, 2013. I believe two common approaches to keep the random structure maximal (random intercepts and random slopes) are: (i) fit the model with the maximal random structure, and stick to that model if it converges, (ii) start with the maximal random structure, and formally compare that model (with AIC/BIC) to models in which you drop one random factor at a time and choose the model with the lowest AIC/BIC.

– Since the authors compare models, I think they were not fitted through restricted maximum likelihood estimation (REML); what method did they use?

– I think it's commendable that the authors considered quadratic models. However, I think they should formally test whether adding a quadratic component significantly improves the fit (via AIC/BIC), and present result accordingly.

– I'm confused about the way "residual pupil size" is calculated. Why is it necessary to first fit a group level model, and then a subject-level? Why not do subject-level directly? Furthermore, it seems very award to not regress out the quadratic relationships here, as we have learned that they are important. I understand the multicollinearity problem, but there may be a more elegant way to deal with this?

I think that the question whether "the effect of baseline pupil state (a peripheral marker of arousal) on perceptual behavior is mediated by the shift in cortical spectral power (a central marker of arousal)" is important. However, I think that the partial correlation approach they use is not ideal to answer this question. Have the authors instead considered to perform an actual mediation analysis? If they want to stick to partial correlation, I think they should do proper partial correlation, and not only regress out cortical spectral power from baseline pupil size, but also from their behavioral metrics, and only then compute the residual relationship between pupil and behavior.

Eyelink records pupil data in arbitrary units. How did the authors meaningfully pool "slow states" across different experimental runs? Z-scoring does not do the trick here. For example, a pupil that is consistently large all the time will result in slow states close to Z=0. Also, strong slow drift results in a large standard deviation, and will push the z-scores down, which is precisely not what you want in that case. Related, why are the units in Figure 3B (z-score) different than in Figure 4C,D (%). In the latter case, % of what?

Are results the same after excluding slow states including blinks? I understand that the authors used ICA to remove blink-related artifacts from the MEG data. However, blinks also cause relatively large transient pupil constrictions (Knapen et al., PLOS ONE, 2016), which will work into the 2-s averages. Related, how often did subjects blink? From Figure 1D it seems about 10 times per minute, which means that 33% of "slow states" might be affected. Finally, why did the authors not use the blink onset and offset timestamps provided by Eyelink? I'm asking because using a constant threshold will miss partial blinks. If the authors want to detect blinks themselves, they should do so based on outliers in the first derivative of the pupil time series.

I think the authors should better integrate their findings with the existing literature, and be more precise with their citations:

– Line 42: "and animals" -> ref. 7 should be cited here as well.

– Line 54: "across a wide range of frequencies" -> ref. 14 did, albeit only in auditory cortex.

– Line 96: "play a functional role in behavior" -> ref. 15, 41 should be cited, and see also de Gee et al., PNAS, 2014; de Gee et al., *eLife*, 2020; Krishnamurthy et al., NHB, 2017; Urai et al., Nat Comm, 2017.

– Line 322: "limited data is available in humans" -> ref. 41 should be cited.

– Line 326: "deserved further investigation" -> ref. 41 should be cited.

*Reviewer #4:*

Podvalny and colleagues investigate the links between spontaneous variations in arousal, as indexed by varying pupil size, and spectral power as well as perceptual decision performance. The study represents extends previous work in non-human animals, initial findings in humans, and improves our understanding of behaviourally relevant changes in arousal and their effects on mesoscopic measures of brain activity (M/EEG). Despite these valuable contributions to the field, central results hinge on methodical approaches that are potentially problematic. In addition to these methodical issues, the transparency and clarity with which results are presented should be improved.

1) Linear mixed effect models:

Although I applaud the general approach of fitting linear mixed effect models to neural and behavioural data, the exact implementation in the current manuscript could be improved and is potentially problematic. First of all, it remains unclear why the authors do not use the potential of LMMs and also fit random slopes to test effects on the single subject level. This is only carried out to partialize pupil size and spectral power, respectively, but not in any models that concern behaviour or the link between spectral power and pupil size. Is there a reason the authors did not fit random slopes? I advocate for the inclusion of random slopes (depending on the model fit) for variables of interest and the inclusion of single subject estimates in all figures. If random slopes for all predictors within the same model lead to convergence issues or problematic model fit, the use of multiple models that only differ in their random slopes (e.g., pupil or spectral power) is acceptable. In this context, it seems to me as if the authors never model behaviour as a joint function of pupil size and spectral power. I strongly suggest to include both spectral power and pupil size in models of behaviour to account for shared variance in the dependent variable of interest (e.g. hit rate or false alarm rate). Furthermore, although the comparison of purely linear and quadratic (including a linear term) models represents an aptly chosen approach, the authors should handle their inferences regarding these model terms with great care and report them more cautiously. Figure 4C for example features visualized quadratic trends that do not represent significant model terms (e.g., sensitivity, response times, etc.). It should be made very clear to the reader that these quadratic effects do not explain additional variance in behaviour and do not fit the data well. In addition, in models where quadratic trends explain significant portions of variance, linear and quadratic effects should be compared directly (e.g., Wald-z statistic) and the results of these comparisons should be reported. Only this way, readers can be able to judge whether linear or quadratic trends are pervasive in a certain relationship. On another note, it is unclear to me if the model coefficients that are presented (e.g., Figure 5) represent standardized betas. If this is the case, they can also work as effect size estimates and an intuitive interpretation should be offered (e.g., "one unit change in pupil size coincides with x…").

2) Stats reporting:

Although the authors have included tables in their manuscript, essential test statistics should be reported within the main text (estimates, confidence intervals, p-values). Despite some p-values, this is missing entirely and should be added. Additionally, I suggest to report standardized effect size estimates to allow for an intuitive interpretation of results. Furthermore, it would be important to note how much variance in brain, behavioural, or pupil data were explained by the winning models. Maybe I have missed this information, if not, it should be added.

3) Stimulus category decoding:

While this is an elegant approach to study the representation of sensory stimuli within neural data and compare them across conditions, the employed chance level is problematic. While it is true that 25% represents the theoretical chance level in a case where four alternatives are discriminated, this is only true for infinite numbers of trials – or at least large datasets (Combrisson et al., 2015, JNeuroMeth). If I understood the approach of bin-wise leave-one out classification correctly, each bin contained about 70 trials. In this case, the empirical chance rate can be calculated as Emp_chance = binoinv(1-α,n_trials,1/n_classes)/n_trials; Where α corresponds to the desired significance level (.05), n_trials to the number of trials and n_classes (4) to the number of possible outcomes. Using this approach, an empirical chance level of 34.2 % is calculated which clearly differs from the theoretical level 0f 25 %. I suggest to either test against the empirical chance level or establish a noise distribution of prediction accuracies using permutations, against which to test.

*Reviewer #5:*

The study by Podvalny et al. measured the relationship between pupil diameter and cortical spectral power, assessing how they provide overlapping versus complementary indices of brain state. The authors used regression analysis to identify relationships between spontaneous, pre-stimulus fluctuations in pupil and MEG spectral power and performance on a visual discrimination task. The analysis revealed correlations between pupil and cortical spectral power but also unique variance between each of these signals and task performance. The experiments and data are novel and are nicely designed and executed. While the results are interesting, they appear in many cases to replicate previous findings. It would help if the authors could more clearly explain the gaps the new results fill in the existing literature.

1. The authors cite work several previous studies in human (Washke et al., Van Kempen et al., de Gee et al) and animal (McGinley et al., Joshi et al., Reimer et al) that explored similar questions and appear to be largely consistent with the current data. This study does appear make a few new comparisons and identifies some differences with the previous work, but the new results appear as a scattered list of observations throughout the discussion. The main observation of explanatory overlap between pupil and cortical spectral power appears to be a replication. It would help if the authors could clarify the novelty of their main results.

2. The modeling results could be presented more clearly. While the general methodology appears sound, it is difficult to discern what aspects of the cortical spectral power are shared with pupil versus not. For example, would it be possible to contrast coefficients for the pupil-independent spectral power model (Figure 5) with coefficients for a model without removing pupil? This information is available in some form in Figures 3 and 4, but it is unclear if and how the units of these different models can be compared. It seems like there should be a unified way to present the joint versus unique contributions of the pupil and MEG signals in terms of variance explained, which would make the results much easier to digest. Without a more comprehensive summary, the results appear as mostly disparate observations.

[Editors’ note: further revisions were suggested prior to acceptance, as described below.]

Thank you for resubmitting your work entitled "Spectral signature and behavioral consequence of spontaneous shifts of pupil-linked arousal in human" for further consideration by *eLife*. Your revised article has been evaluated by Joshua Gold (Senior Editor), a Reviewing Editor, the original reviewers, and a new reviewer.

This study takes a comprehensive look at the relationship between pupil-indexed arousal and cortical MEG activity and performance on a demanding sensory behavior in humans subjects. Understanding how pupil activity, widely interpreted as a marker of arousal, relates to cortical activity, is a key challenge in systems and cognitive neuroscience. The authors identified a widespread component of cortical activity that can be explained by changes in pupil and that predict performance on a behavioral task. This work helps build an important bridge between animal models of arousal and human cognitive behavior.

Essential revisions:

Presentation of single subject data (as per Rev #1).

1) Presentation of effect sizes and explained variance (as per Rev #1).

2) Are pupil-related effects really essentially uniform across the cortex? Or are the differences really the important thing? It would be nice if this could be addressed even briefly in the Discussion. (as per Rev #3)

3) Since the present analysis is focused on quantifying pupil size it is important to account for other factors which might affect measured pupil size, notably, fixation and saccades. The section devoted to detailing the analysis of the pupil data omits any discussion of these factors. Did the authors check that fixation is maintained at all times? Were periods associated with saccades away from fixation (which will result in apparent changes to pupil size) removed from the analysis? (as per Rev #4)

4) The analysis of "fast events" was not very clear to us (as per Rev #4, please follow the details for an additional/re-analysis there):

a) we are worried about the low-pass filter and a potential acausal smearing it might have contributed.

b) The statement (line 492) "to be considered a peak/through, the data point should have highest/lowest amplitude within 500 ms vicinity" was also not clear. It would be helpful if you plotted (or reported) the duration distribution of the events you ended up analyzing (as per Rev #4, see details also below).

*Reviewer #1:*

The authors have revised large parts of their manuscript and addressed most comments.

However, two of my previous concerns have not been resolved entirely.

1) Presentation of single subject data.

The authors argue that this is not feasible as it would require to plot more than 8000 data points. This likely is a misunderstanding. While it is true that plotting single subject points for all power / frequency / ROI bins is not feasible, this is not what I asked for. Take figure 3B as an example. In every panel, the authors plot spectral power (in bins) as a function of pupil size (in bins). They plot one data point per bin (with an error bar around it). If I am not entirely mistaken, this data point represents the group average. It is essential, however, to show single subject data at this point. What speaks against plotting a cloud of N=24 dots around their mean instead of each dot? I realize that this might take up a bit more space but I am convinced that this is an essential service to every reader. Importantly, the same comment applies to figure panels 4C, 4D, and 4E.

2) Effect sizes and explained variance

While it is true that the concept of variance is explained is not easily transferred to the world of linear mixed models, it is entirely possible to offer such an estimate, both for linear mixed models as well as generalized mixed models. Even if imperfect, such an approximation represents an important piece of information since it puts the models in perspective to the space they were trying to capture. I allow myself to recommend the excellent R-package lme4 which allows the fitting of such models and in combination with functions from the sj_plot package (e.g., tab_model) offers all relevant information. I am sure transferring their models from python to R should be a simple task for the authors.

Furthermore, I suggest to offer an explicit discussion of model fits and effect sizes which is missing in the current version of the manuscript. The supplementary tables illustrate that effects rather are small and likely don't account for large parts of variance in the data. This does not have to be an issue but it should be acknowledged prominently and discussed.

*Reviewer #2:*

The authors have performed a thorough revision and addressed my concerns. I have no remaining issues.

*Reviewer #3:*The authors were responsive to all the concerns raised in the initial review. The revised manuscript clarifies the significance of the work and more rigorously identifies the link between pupil size, cortical activity and task performance.Overall, the manuscript looks great. Just a small lingering question. The effects in Figure 3 appear strikingly consistent across brain networks, and any differences appear to be quantitative more than qualitative (ie, even where a quadratic effect may not be significant, one can still see a trend that is consistent with networks showing a quadratic effect). Does this mean that pupil-related effects are more or less uniform across the cortex? Or are the differences really the important thing? It would be nice if this could be addressed even briefly in the Discussion.

*Reviewer #4:*

I joined the review process at the revision stage. I did not read the previous version of the manuscript or the rebuttal later in order to form an unbiased view of the work.

1) The experiment is quite long (over an hour and a half). In addition to task related effort, experimental length can have consequences for pupil dynamics due to effort related to maintain fixation and accommodation. It is critical that the authors demonstrate that pre-stimulus pupil size is not correlated with time within the experiment.

2) Similarly, since the present analysis is focused on quantifying pupil size it is important to account for other factors which might affect measured pupil size, notably, fixation and saccades. The section devoted to detailing the analysis of the pupil data omits any discussion of these factors. Did the authors check that fixation is maintained at all times? Were periods associated with saccades away from fixation (which will result in apparent changes to pupil size) removed from the analysis?

3) The analysis of "fast events" was not very clear to me.

a) Why was a low pass filter applied to the data before identifying the phasic events? Wouldn't a low pass filter smear the timing of these events? Depending on the filter (no details are provided) this could result in latency shifts on the order of 200ms, making the "fast event" analysis (Figure 6) difficult to interpret.

b) The statement (line 492) "to be considered a peak/through, the data point should have highest/lowest amplitude within 500 ms vicinity" was also not clear. It would be helpful if you plotted (or reported) the duration distribution of the events you ended up analyzing and maybe follow the analysis approach as used in e.g. Joshi et al. (2016; Neuron) where phasic events are defined as zero-crossings of the pupil slope separated by {greater than or equal to}75 ms. This will make it easier to relate your findings to the literature on phasic pupil responses.

4) Analysis in Figure 6 (and page 12).

a) Why was a Pearson and not spearman correlation used? The latter is less sensitive to outliers and would provide stronger evidence for the claims the authors wish to make.

b) Panel C: I am a bit confused by the example of identified pupil events provided. In the methods you state that you defined dilation/constriction events as having a duration of at most 500 ms. But from eye balling the figure the first dilation event (left most blue dot) appears to last more than a second? (Similarly the right most blue dot in the plot).

c) I would like to see more discussion of the shape of the patter of MEG activity triggered by the pupil events. I found the pattern extremely surprising. e.g. that the sharp peak/trough exactly coincides with 0; is it possible that the preceding peak at ~-400 relates to a previous dilation/constriction event? I would appreciate more detailed discussion of what this all means. Similarly the pattern of correlation across channels looks quite systematic. Did the authors try to source localize this pattern?

Other points

1) It is stated that sensitivity (d') is related to prestimulus size in an inverted U-shaped relationship. This is not obvious at all from looking at the data points in the figure. Instead the behavioral performance appears to not differ much for pupil bins 2-5.

2) I understand it is difficult to control in the present experiment, but doesn't a larger pupil also imply stronger visual input? Would this be able to explain the largely linear link between pupil size and brain activity in the β and γ ranges?

---

## [Author Response]

[Editors’ note: the authors resubmitted a revised version of the paper for consideration. What follows is the authors’ response to the first round of review.]

Reviewer #3:I think this is an important paper. The characterization of the relationship between baseline pupil size and large-scale cortical spectral power across a wide range of frequencies (localized using Dynamic Imaging of Coherent Sources) is state of the art and novel. However, I do have a number of serious issues with the methods. I expect at least some of the results will change when the authors address these.

We thank the reviewer for recognizing the importance of our study. The methodological issues raised by the reviewer have now been fully addressed, as explained below.

I have a number of issues about the mixed effects modeling.– The authors write that they only considered random intercepts, and not random slopes. Why? See for example Barr et al., Journal of Memory and Language, 2013. I believe two common approaches to keep the random structure maximal (random intercepts and random slopes) are: (i) fit the model with the maximal random structure, and stick to that model if it converges, (ii) start with the maximal random structure, and formally compare that model (with AIC/BIC) to models in which you drop one random factor at a time and choose the model with the lowest AIC/BIC.

We now followed suggestion (i) by the reviewer – the models converged in the vast majority of cases.

The methods section on “Statistical modeling” (PP. 20-21) has been thoroughly updated. E.g., we explain:

“We used LMMs with the maximal random effects structure justified by the design (Barr et al., 2013).”

The equations for the LMM models (Equation 3-6) have been expanded to include the random intercepts (ℓ_!"_), random slopes (ℓ_#"_) and random-effects quadratic coefficients (ℓ_$"_). In Equation 3-4, because spectral power was demeaned before the analysis, random intercepts were 0 and thus omitted from the equation.

Figure 3B-C, Figure 4C-D, Figure 5, Supplementary figure 1, Table 1, Supplementary tables 1-2 have been updated along with the corresponding Results sections.

The results remained qualitatively similar: first, spontaneous fluctuations in pupil size were significantly associated with power fluctuations in most resting state networks and frequency bands (Figure 3B-C); second, spontaneous fluctuations in pupil size before stimulus onset predicted behavior (Figure 4C-D); third, residual power unpredicted by pupil-linked arousal had additional effects on behavior (Figure 5).

– Since the authors compare models, I think they were not fitted through restricted maximum likelihood estimation (REML); what method did they use?

We now use the standard (not restricted) likelihood in all cases, as clarified in Methods (lines 541-542):

“The models were fit using Powell’s algorithm (Powell, 1964) minimizing standard (rather than restricted) likelihood to assure the meaningful information criterion calculation”.

– I think it's commendable that the authors considered quadratic models. However, I think they should formally test whether adding a quadratic component significantly improves the fit (via AIC/BIC), and present result accordingly.

Following the reviewer’s suggestion, in Figure 3B-C and Supplementary Figure 1, we now only present the winning models. That is, models with quadratic components are now only plotted if their BIC values are lower than the corresponding linear models (lines 124-129):

“We fit two types of models, one with both linear and quadratic components (Equation 3, Methods), […] and one with a linear component only (Equation 4, Methods) for comparison. Figure 3B presents the fitted curves from the task baseline data (similar plots from rest are shown in Supplementary Figure 1) for the models with lower BIC, and Figure 3C presents the linear parameter estimates for both task baseline and rest, and a quadratic parameter estimate in case the model that included this parameter was preferred according to BIC” .

In Figure 4C-D, to facilitate comparison with earlier studies that have reported both linear and quadratic relationships between pupil size and perceptual behavior, we present both models with and without the quadratic component as long as the parameter estimates are significant. We further report the full statistics for each model and their BIC values in Table 1 and in lines 186-188:

“Previous studies reported linear and/or quadric models fitting such behavioral metrics and task-evoked pupil responses(de Gee et al., 2017; McGinley et al., 2015; Waschke et al., 2019) and, accordingly, we also report the results of both model types to allow comparison (see Methods, Equation 5-6).”

– I'm confused about the way "residual pupil size" is calculated. Why is it necessary to first fit a group level model, and then a subject-level? Why not do subject-level directly? Furthermore, it seems very award to not regress out the quadratic relationships here, as we have learned that they are important. I understand the multicollinearity problem, but there may be a more elegant way to deal with this?

We originally fit the model on the group level to select a set of regressors which subsequently would be shared by the individual subjects to increase interpretability and reproducibility. The selection process was needed to attempt to solve the multicollinearity problem because the whole-brain model including all RSNs and frequency bands produced a large number of correlated regressors. However, because of the issue with quadratic relationships raised by the reviewer, we have decided to remove this analysis (originally presented in Supplementary figure 4) and instead introduce a new analysis testing whether pupil size contains additional information that contributes to the prediction of “yes”/”no” reports on a single-trial level (new Figure 5A, and see more details in the point below). In this new analysis, residual power was calculated by using either the linear or the quadratic model—selected for individual subjects based on BIC (explained on lines 566-569).

I think that the question whether "the effect of baseline pupil state (a peripheral marker of arousal) on perceptual behavior is mediated by the shift in cortical spectral power (a central marker of arousal)" is important. However, I think that the partial correlation approach they use is not ideal to answer this question. Have the authors instead considered to perform an actual mediation analysis? If they want to stick to partial correlation, I think they should do proper partial correlation, and not only regress out cortical spectral power from baseline pupil size, but also from their behavioral metrics, and only then compute the residual relationship between pupil and behavior.

Because the behavioral metrics (e.g., d’, c, hit rate and false alarm rates) are calculated on groups of trials (instead of single trials), it is unfortunately not possible to regress out the cortical power from behavioral metrics as the reviewer suggested. We have now removed the analysis of residual pupil-linked arousal and included a new analysis testing whether pupil size contains additional information that contributes to the prediction of “yes”/”no” reports on a single-trial level beyond information contained in cortical spectral power (new Figure 5A; described in lines 218-231):

“To understand the respective contributions of spectral power that can be explained by pupil-linked arousal (Figures 2-3) and of spectral power independent of arousal to perceptual behavior, we first fit logistic regression models predicting “yes”/”no” recognition behavior from baseline fluctuations of spectral power in all frequency bands and RSNs only (“power only”), models with pupil size in addition (“pupil and power”), models using residual power independent of pupil size (“residual power”) and models with pupil size as a single feature (“pupil only”) (Figure 5A, see Methods). The residual power was calculated by first fitting LMM models for each frequency band and each RSN for each participant, whereby the model with lower BIC at the group level was chosen between Equation 3 and Equation 4 (see Methods, Table 1). We find that all logistic regression models predict “yes”/”no” behavior better than chance (permutation test, p<0.05 FDR corrected). Interestingly, the performance of “power only” model does not improve by adding pupil size information (W = 143, p = 0.86, Wilcoxon signed-rank test), but the performance is worsened by regressing out the pupil-linked spectral power (W = 241, p = 0.008, Wilcoxon signed-rank test). These results indicate that both arousal-linked and arousal-independent spectral power fluctuations at baseline predict subsequent recognition behavior, but our recordings of pupil size did not provide additional information contributing to the prediction of behavior that was not already present in cortical spectral power.”

This new analysis confirms our main conclusion that both arousal-linked and arousal-independent spectral power fluctuations at baseline predict subsequent recognition behavior.

Eyelink records pupil data in arbitrary units. How did the authors meaningfully pool "slow states" across different experimental runs? Z-scoring does not do the trick here. For example, a pupil that is consistently large all the time will result in slow states close to Z=0. Also, strong slow drift results in a large standard deviation, and will push the z-scores down, which is precisely not what you want in that case. Related, why are the units in Figure 3B (z-score) different than in Figure 4C,D (%). In the latter case, % of what?

The reviewer is correct in that the timescales longer than an experiment run could not be assessed in the present study, we now discuss this limitation in lines 413-417:

“…spontaneous fluctuations in pupil size on a timescale longer than a run length (5.6 ± 0.38 min) might be important. In the present experiment such longer time scales could not be assessed since we used a relative measure of pupil size change within each experiment block, whereas an absolute pupil size measure (not available using the present eye-tracking set-up) is necessary to assess change in pupil size across blocks.”

We note that “slow” and “fast” are relative terminology. The fast pupil changes investigated herein are at the time scale of tens to hundreds of milliseconds (Figure 6). By contrast, for the slow states, we average pupil size in 2-sec time windows, which corresponds to a <0.5 Hz filter. Therefore, our ‘slow states’ reflect pupil size changes in the range of 0.003–0.5 Hz. We note that our method to assess slow states, involving normalization (z-scoring) within a block, is very similar to resting-state fMRI studies. fMRI also does not provide measures of BOLD signal in absolute unit; therefore, it is customary to normalize the fMRI signal per experimental run (~5 min) to change it into a z-score or %change unit. Similar to the present study, resting-state fMRI research is unable to assess slow changes at timescales beyond that of an experimental run, and this has not prevented that field from generating a wealth of insightful findings about slow fMRI activity fluctuations. Z-scoring per run was also used in previous pupil studies (e.g., Knapen et al., 2016 referred by the reviewer, which had 2–2.5 min runs).

Lastly, we note that the relationship between the ‘slow states’ investigated herein (at 0.003–0.5 Hz range) and the ‘tonic pupil size’ investigated in previous studies such as Knapen et al., 2016 (at <0.02 Hz range) requires additional investigation; in both studies, relatively arbitrary frequency cut-offs were used for these definitions.

The unit in Figure 3B is z-score, which reflects per-block normalized pupil size fluctuations averaged in 2sec prestimulus windows (i.e., slow states), which were used to fit the models relating to MEG power using single-trial data. For Figure 4C-D the trials were binned into five groups according to quintiles of the 2-sec averaged prestimulus pupils size in order to calculate behavioral metrics (such as d’ and c). We have now clarified that the x-axis in Figure 4C-D is pupil size bin, defined according to pupil size quintiles.

Are results the same after excluding slow states including blinks? I understand that the authors used ICA to remove blink-related artifacts from the MEG data. However, blinks also cause relatively large transient pupil constrictions (Knapen et al., PLOS ONE, 2016), which will work into the 2-s averages.

Given that normal spontaneous blink rate can reach 20/min and the blink-related transients last for up to 4 seconds (Knapen et al., 2016), excluding data would mean throwing away most experiment trials. We now discuss this limitation in lines 421-429:

“…the full composition of endogenous events driving spontaneous fluctuations in pupil size remains incompletely understood. While a causal association between activity in brainstem neuromodulatory centers and pupil size has been established, additional endogenous factors, ranging from cognitive (e.g., an exciting thought) to physiological (e.g., blinks), affect the pupil size. For example, it has been shown that blinks correlate with a subsequent fluctuation in pupil size lasting for up to 4 seconds (Knapen et al., 2016). Importantly, it is unclear if such endogenous effect can be dissociated from arousal or not. Given such a long-lasting effect, excluding trials with blinks within 4-sec intervals is not feasible (normal human blink rate can reach ~1/3 sec). Thus, future research is needed to determine to what extent different endogenous events drive the spontaneous fluctuations in pupil size in addition to the known neuromodulatory effects.”

Related, how often did subjects blink? From Figure 1D it seems about 10 times per minute, which means that 33% of "slow states" might be affected.

The blink rate is 17.62 ± 1.77 times per minute in our data. Note that our relatively liberal criterion for blink definition does not dissociate between the potential loss of tracking and blinks (explained in the comment below), hence the actual blink rate could be lower. We instructed the subjects to try and blink after the stimulus presentation and avoid blinking during the prestimulus interval whenever possible, in order to better assess how prestimulus baseline pupil size varies as a function of cortical spectral power (Figures 2-3) and subsequent perceptual behavior (Figure 4). In our data, only 33.08 ± 4.6% of 2-sec prestimulus intervals analyzed were affected by blinks.

Finally, why did the authors not use the blink onset and offset timestamps provided by Eyelink? I'm asking because using a constant threshold will miss partial blinks. If the authors want to detect blinks themselves, they should do so based on outliers in the first derivative of the pupil time series.

The reviewer is correct in that our procedure might miss incomplete blinks or partial eye closures. Our understanding is that EyeLink’s detection procedure would miss the incomplete blinks as well. EyeLink blink detection code is proprietary and is not publicly available; however, from reading the User Manual (page 24), it seems that to identify an event as a blink the pupil data must be missing: “A blink is defined as a period of saccade detector activity with the pupil data missing for three or more samples in a sequence.” Our understanding is that such a procedure as implemented in Eyelink will not identify events as blinks when the pupil data is not missing at all, as in the case of partial eye closure.

Without using an additional criterion of velocity change, our procedure will identify all events with missing data, wherein the missing data could be due to eye closure or due to lost tacking, as “blinks”. This was intended since we do want to exclude the instances of lost tracking (instead of using interpolation) to be conservative, but it might slightly increase the identified blink rate. We have now clarified this in methods (lines 483-485, new text underlined):

“Blinks or missing data periods were detected by identifying the time points where the recorded pupil size of the right eye dropped below a constant threshold”

I think the authors should better integrate their findings with the existing literature, and be more precise with their citations:– Line 42: "and animals" -> ref. 7 should be cited here as well.

Done.

– Line 54: "across a wide range of frequencies" -> ref. 14 did, albeit only in auditory cortex.

The full sentence including the clause mentioned above is: “Importantly, previous studies in this domain have not investigated how pupil-linked arousal covaries with large-scale cortical spectral power across a wide range of frequencies”. Our understanding is that ref. 14 did not investigate large-scale cortical spectral power.

– Line 96: "play a functional role in behavior" -> ref. 15, 41 should be cited, and see also de Gee et al., PNAS, 2014; de Gee et al., eLife, 2020; Krishnamurthy et al., NHB, 2017; Urai et al., Nat Comm, 2017.

The sentence including the clause mentioned above began with: *“While baseline pupil diameter …”*, to clarify that we specifically refer to spontaneous (non-stimulus-triggered) dilation and constrictions, we edited the sentence as follow: (lines 95-97, new text underlined)

“While baseline pupil diameter (“slow state” equivalent) correlates with subsequent perceptual detection in humans (Podvalny et al., 2019; Van Kempen et al., 2019; Waschke et al., 2019) and mice (Aston-Jones and Cohen, 2005; McGinley et al., 2015; Steinmetz et al., 2019), it is currently unknown whether momentary spontaneous dilations and constriction (“fast events”) during a baseline period play a functional role in behavior.”

The papers mentioned by the reviewer investigated stimulus-triggered pupil responses and therefore were not cited here in particular.

– Line 322: "limited data is available in humans" -> ref. 41 should be cited

Done (though we again refer to spontaneous activity).

– Line 326: "deserved further investigation" -> ref. 41 should be cited

Our understanding is that *“spontaneous pupil size fluctuations”* were not investigated in 41. No doubt Ref 41 is an important paper, but we are afraid that citing it here in the context of spontaneous fluctuations could cause confusion for the reader.

Reviewer #4:Podvalny and colleagues investigate the links between spontaneous variations in arousal, as indexed by varying pupil size, and spectral power as well as perceptual decision performance. The study represents extends previous work in non-human animals, initial findings in humans, and improves our understanding of behaviourally relevant changes in arousal and their effects on mesoscopic measures of brain activity (M/EEG). Despite these valuable contributions to the field, central results hinge on methodical approaches that are potentially problematic. In addition to these methodical issues, the transparency and clarity with which results are presented should be improved.

We thank the reviewer for recognizing the value of these contributions. The methodological issues identified by the reviewer have been fully addressed, and the transparency and clarity of the presentation has been enhanced.

1) Linear mixed effect models:Although I applaud the general approach of fitting linear mixed effect models to neural and behavioural data, the exact implementation in the current manuscript could be improved and is potentially problematic. First of all, it remains unclear why the authors do not use the potential of LMMs and also fit random slopes to test effects on the single subject level. This is only carried out to partialize pupil size and spectral power, respectively, but not in any models that concern behaviour or the link between spectral power and pupil size. Is there a reason the authors did not fit random slopes? I advocate for the inclusion of random slopes (depending on the model fit) for variables of interest and the inclusion of single subject estimates in all figures. If random slopes for all predictors within the same model lead to convergence issues or problematic model fit, the use of multiple models that only differ in their random slopes (e.g., pupil or spectral power) is acceptable.

We have now included random slopes in all LMM analyses in the present study to keep the random structure maximal (please also see our response to Reviewer #3’s point 1 above). We added Supplementary figure 3 showing individual-subject data and model fit for analyses presented in Figure 4C-D (including radom slopes and random intercepts). We did not include individual subject data and model fit for analyses presented in Figure 3B since they include 8460 data points for each frequency band and RSN (the visualization in Figure 3B uses 5% pupil-size bins) and it is difficult to obtain a useful visual representation of such extensive data at the individual-subject level.

In this context, it seems to me as if the authors never model behaviour as a joint function of pupil size and spectral power. I strongly suggest to include both spectral power and pupil size in models of behaviour to account for shared variance in the dependent variable of interest (e.g. hit rate or false alarm rate).

To calculate the behavioral variables according to Signal Detection Theory we had to split the trials in groups. These behavioral variables (HR, FAR, d’, c) cannot be computed at the single-trial level. For this reason, we split the trials into groups according to an independent variable (i.e., prestimulus spectral power or pupil size) and test whether such grouping predicts behavioral changes (dependent variable). As far as we understand, it is not possible to define groups of trials and calculate SDT metrics according to more than one independent variables at the same time.

However, to address the reviewer concern, we have now introduced an additional single-trial level analysis of a model predicting behavioral responses (“yes”/”no”) using both prestimulus pupil size and spectral power, as the reviewer suggested. This analysis allowed us to test whether pupil size contains additional information that contributes to the prediction of “yes”/”no” reports on a single-trial level beyond information contained in cortical spectral power (reported in new Figure 5A; described in lines 218231):

“To understand the respective contributions of spectral power that can be explained by pupil-linked arousal (Figures 2-3) and of spectral power independent of arousal to perceptual behavior, we first fit logistic regression models predicting “yes”/”no” recognition behavior from baseline fluctuations of spectral power in all frequency bands and RSNs only (“power only”), models with pupil size in addition (“pupil and power”), models using residual power independent of pupil size (“residual power”) and models with pupil size as a single feature (“pupil only”) (Figure 5A, see Methods). The residual power was calculated by first fitting LMM models for each frequency band and each RSN for each participant, whereby the model with lower BIC at the group level was chosen between Equation 3 and Equation 4 (see Methods, Table 1). We find that all logistic regression models predict “yes”/”no” behavior better than chance (permutation test, p<0.05 FDR corrected). Interestingly, the performance of “power only” model does not improve by adding pupil size information (W = 143, p = 0.86, Wilcoxon signed-rank test), but the performance is worsened by regressing out the pupil-linked spectral power (W = 241, p = 0.008, Wilcoxon signed-rank test). These results indicate that both arousal-linked and arousal-independent spectral power fluctuations at baseline predict subsequent recognition behavior, but our recordings of pupil size did not provide additional information contributing to the prediction of behavior that was not already present in cortical spectral power.”

This new analysis confirms our main conclusion that both arousal-linked and arousal-independent spectral power fluctuations at baseline predict subsequent recognition behavior.

Furthermore, although the comparison of purely linear and quadratic (including a linear term) models represents an aptly chosen approach, the authors should handle their inferences regarding these model terms with great care and report them more cautiously. Figure 4C for example features visualized quadratic trends that do not represent significant model terms (e.g., sensitivity, response times, etc.). It should be made very clear to the reader that these quadratic effects do not explain additional variance in behaviour and do not fit the data well.

We thank the reviewer for raising this issue. We have now updated Figure 4C-D to include only models with significant parameter estimates. As such, the quadratic models for categorization behavior (including reaction times and accuracy in Figure 4D) have now been removed.

In addition, in models where quadratric trends explain significant portions of variance, linear and quadratic effects should be compared directly (e.g., Wald-z statistic) and the results of these comparisons should be reported. Only this way, readers can be able to judge whether linear or quadratic trends are pervasive in a certain relationship.

We have used Bayesian information criterion (BIC) to select a better model between the linear and quadratic models. In response to Reviewer #3’s point 3, in Figure 3B-C and Supplementary figure 1, we now only present the winning models. That is, models with quadratic components are now only plotted if their BIC values are lower than the corresponding linear models. In Figure 4C-D, to facilitate comparison with earlier studies that have reported both linear and quadratic relationships between pupil size and perceptual behavior, we present both models with and without the quadratic component as long as the quadratic component is significant (also see our response to the point above). We further report the full statistics for each model and their BIC values in Table 1.

On another note, it is unclear to me if the model coefficients that are presented (e.g., Figure 5) represent standardized betas. If this is the case, they can also work as effect size estimates and an intuitive interpretation should be offered (e.g., "one unit change in pupil size coincides with x…").

All reported coefficients are standardized now:

“Pupil size was z-scored to produce standardized effect size” (lines 553)

“The group numbers were z-scored to produce standardized effect size” (line 565)

2) Stats reporting:Although the authors have included tables in their manuscript, essential test statistics should be reported within the main text (estimates, confidence intervals, p-values). Despite some p-values, this is missing entirely and should be added.

We report the full statistics, including coefficient estimates, standard deviations, BIC values, for all tested models (including 70 models in Figure 3 and 210 models in Figure 5, both significant and non-significant) in supplementary tables. Due to the large number of models tested, even if we only restrict the statistical reporting to significant models in the main text, including the full statistics would be impractical—the sheer length of statistical reporting would severely impair readability of the text. Our SI tables are well organized such that it is easy for the reader to find the full statistics for any of the tested models.

Additionally, I suggest to report standardized effect size estimates to allow for an intuitive interpretation of results.

Done.

Furthermore, it would be important to note how much variance in brain, behavioural, or pupil data were explained by the winning models. Maybe I have missed this information, if not, it should be added.

While the concept of ‘variance explained’ is indeed intuitive and is very often used as a summary statistic to quantify the goodness-of-fit of fixed-effects models (e.g., linear regressions, ANOVA, or generalized linear models (GLMs)), generalization of R^2^ to LMM is challenging. There are some theoretical developments on this topic (e.g., Nakagawa and Schielzeth, 2013), but these methods are not yet implemented in the software package we are using (*statsmodels* python toolbox, version 0.12.2).

3) Stimulus category decoding:While this is an elegant approach to study the representation of sensory stimuli within neural data and compare them across conditions, the employed chance level is problematic. While it is true that 25% represents the theoretical chance level in a case where four alternatives are discriminated, this is only true for infinite numbers of trials – or at least large datasets (Combrisson et al., 2015, JNeuroMeth). If I understood the approach of bin-wise leave-one out classification correctly, each bin contained about 70 trials. In this case, the empirical chance rate can be calculated as Emp_chance = binoinv(1-α,n_trials,1/n_classes)/n_trials; Where α corresponds to the desired significance level (.05), n_trials to the number of trials and n_classes (4) to the number of possible outcomes. Using this approach, an empirical chance level of 34.2 % is calculated which clearly differs from the theoretical level 0f 25 %. I suggest to either test against the empirical chance level or establish a noise distribution of prediction accuracies using permutations, against which to test.

In the previous version of the manuscript we did not address the question of whether the decoding accuracy is above chance within a specific group of trials. Instead, we were interested in the question of whether the prestimulus pupil size has an effect on subsequent neural representation of the stimulus. We realize, however, it is important to show the empirical (or permutation-derived) chance level and to assess whether the decoding accuracy is significantly higher than that. We now address this question by performing 500 permutations with shuffled object category labels within each bin, and derived the permutation-established chance level, which was 0.246 ± 0.06 (mean ± sd). We now added the permutation-derived chance level and decoding significance (computed using the 500-permutation null distribution) to Figure 4E and we report the results in text (lines 211-214):

“In addition, the decoding accuracy was significantly above the chance level (obtained by label permutations) from 200 ms to 1 s after stimulus onset (p < 0.05, FDR corrected) and significantly above the chance level for trials in the second, forth, and fifth pupil size groups.”

Reviewer #5:The study by Podvalny et al. measured the relationship between pupil diameter and cortical spectral power, assessing how they provide overlapping versus complementary indices of brain state. The authors used regression analysis to identify relationships between spontaneous, pre-stimulus fluctuations in pupil and MEG spectral power and performance on a visual discrimination task. The analysis revealed correlations between pupil and cortical spectral power but also unique variance between each of these signals and task performance. The experiments and data are novel and are nicely designed and executed. While the results are interesting, they appear in many cases to replicate previous findings. It would help if the authors could more clearly explain the gaps the new results fill in the existing literature.

We thank the reviewer for appreciating our study design and analysis, and the novelty thereof.

1. The authors cite work several previous studies in human (Washke et al., Van Kempen et al., de Gee et al) and animal (McGinley et al., Joshi et al., Reimer et al) that explored similar questions and appear to be largely consistent with the current data. This study does appear make a few new comparisons and identifies some differences with the previous work, but the new results appear as a scattered list of observations throughout the discussion. The main observation of explanatory overlap between pupil and cortical spectral power appears to be a replication. It would help if the authors could clarify the novelty of their main results.

Our study contributes several novel results and below we address the reviewer’s concern in the context of the previous studies mentioned:

First, we provide the characterization of the relationship between the spectral power of neural activity localized in large-scale cortical networks during *spontaneous* fluctuations of pupil-linked arousal at rest and prestimulus baseline. While the results of this analysis are mostly, but not always, consistent with prior studies in humans and animals, the scale of this analysis and the precise characterization of the pupil-brain relationship are novel. Waschke et al. 2019 described pupil-brain relationships before stimulus onset using EEG activity in the auditory cortex only, for example. Furthermore, our study is the first study using MEG, which provides superior signal-to-noise and source localization as compared to EEG. This may explain why some previous studies reported null effects, such as no relationship between α power and pupil size reported in Waschke et al. 2019, while we found a significant inverted-U relationship. While our results are consistent with prior findings from animal electrophysiology, these animal studies have recorded a small number of brain regions (auditory cortex in McGinley et al.; LC in Joshi et al; V1 and S1 in Reimer et al.); as such, the large-scale distribution of pupil-power relationship across the whole cortex was not known.

Second, we report the effect of pupil-linked arousal fluctuations *at baseline* on behavior in visual perceptual decision-making task. Our study was specifically designed to examine prestimulus baseline activity by utilizing longer prestimulus intervals than in previous studies, variable stimulus presentation times to reduce the stimulus expectation effects, and a lack of any additional task-relevant stimuli present during baseline. Van Kempen et al. 2019 and de Gee et al. 2017, for example, defined baseline while a non-target stimulus was in fact present on the screen, which may mask the spontaneous arousal fluctuations. Our study design also allowed us to study changes in multiple metrics of behavior, whereas previous studies, such as Van Kempen et al. 2019, reported the effect of pre-target baseline pupil size on reaction times only (accuracy was at ceiling). In sum, our study was designed to provide a more complete picture specifically on the topic of spontaneous arousal fluctuations and their role in perceptual decision making under uncertainty.

Third, in the present work we delineate the specific behavioral effects of pupil-linked and pupil independent spectral power fluctuations at baseline. Multiple previous studies explored spectral power within one limited frequency band (usually α) localized to one brain area without controlling for arousal. In the present study we study this question using MEG (better SNR, better source localization) and controlling for arousal. We find that a significant portion of spectral power’s effect on behavior is arousal mediated and but arousal-independent cortical activity power fluctuations also contribute significantly to perceptual behavior. Partitioning cortical activity power’s influence on perceptual behavior into components that are arousal-linked and arousal-independent is a novel step undertaken in this study.

Finally, our study is the first to begin to identify the large-scale neural mechanisms of spontaneous pupil linked arousal *on a faster time-scale in human*. We show that large-scale brain activity correlates with pupil size on a millisecond time scale in human. While a similar analysis was conducted in animals using intracranial electrophysiology, a time lag of 400ms that we found in human, considering the monkey LC time lag is 300 ms (Joshi et al. 2016, brain activity preceding the pupil size), calls for investigation of the feedback loops from cortex to LC as a mechanism of arousal control during baseline and rest. Furthermore, we found that spontaneous pupillary events in human at this time scale did not correlate with subsequent visual perceptual decision-making. Such events and their neural correlates were studied in animals (Reimer et al. 2014) but their behavioral relevance was not tested.

In sum, our study makes several novel contributions to the field, which were summarized very well in the editors’ decision letter: “The reviewers and editors felt that this manuscript aims at offering important insights. First, brain-wide relationships across a wide range of spectra have only rarely been tested in humans. Second, testing direct and indirect links between baseline pupil size, spectral power and behaviour, separately, has not been done before in humans, to the best of our knowledge. Finally, the authors try to combine and compare the behavioural relevance of slow as well as fast pupil size fluctuations, linking to previous research in non-human animals.” The decision letter further specified that the main issue with our previous submission was methodological, which we have fully addressed now.

2. The modeling results could be presented more clearly. While the general methodology appears sound, it is difficult to discern what aspects of the cortical spectral power are shared with pupil versus not. For example, would it be possible to contrast coefficients for the pupil-independent spectral power model (Figure 5) with coefficients for a model without removing pupil? This information is available in some form in Figures 3 and 4, but it is unclear if and how the units of these different models can be compared. It seems like there should be a unified way to present the joint versus unique contributions of the pupil and MEG signals in terms of variance explained, which would make the results much easier to digest. Without a more comprehensive summary, the results appear as mostly disparate observations.

While the concept of ‘variance explained’ is indeed intuitive and is often used as a summary statistic to quantify the goodness-of-fit of fixed-effects models (e.g., linear regressions, ANOVA, or generalized linear models), generalization of R^2^ to LMM is challenging. There are some theoretical developments on this topic (e.g., Nakagawa and Schielzeth, 2013), but these methods are not yet implemented in the software package we are using (*statsmodels* python toolbox, version 0.12.2). However, we have implemented a new analysis to address this concern, as described below.

The reviewer inquired about whether it is possible to compare the regression coefficients of two models, one predicting behavior with pupil-independent power and one with full power. In the linear case of a model for criterion (c), such analysis would constitute a comparison between the coefficients *X_L_*_1_ and *X_L_*_2_ of the models below:

C1∼XL1∙Group_power+XO1+ε1

 C2∼XL2∙Group_residential_power+XO1+ε1

Please note that our behavioral metrics here cannot be computed on a single-trial level (i.e., one needs to analyze a group of trials to estimate criterion), and they vary with prestimulus states, which were defined differently for the two models. For these reasons, we cannot draw conclusions about the differences between the coefficients of the two models, as this would call for a single model explaining behavior with both full and residual power and testing the interaction between these terms, which is not possible in this case.

However, to address the reviewer’s concern, we have now implemented Logistic regression models predicting subjects’ recognition behavior (“yes”/”no”) on a single-trial level, using baseline cortical spectral power alone, baseline pupil size alone, power plus pupil, and pupil-independent residual power. The new analysis is reported in the new Figure 5A and described in lines 218-231. It was also referred to in response to Reviewer #3’s point 2 and Reviewer #4’s point 1.

“To understand the respective contributions of spectral power that can be explained by pupil-linked arousal (Figures 2-3) and of spectral power independent of arousal to perceptual behavior, we first fit logistic regression models predicting “yes”/”no” recognition behavior from baseline fluctuations of spectral power in all frequency bands and RSNs (“power only”), models with pupil size in addition (“pupil and power”), models using residual power independent of pupil size (“residual power”) and models with pupil size as a single feature (“pupil only”) (Figure 5A, see Methods). The residual power was calculated by first fitting LMM models for each frequency band and each RSN for each participant, whereby the model with lower BIC at the group level was chosen between Equation 3 and Equation 4 (see Methods, Table 1). We find that all logistic regression models predict “yes”/”no” behavior better than chance (permutation test, p<0.05 FDR corrected). Interestingly, the performance of “power only” model does not improve by adding pupil size information (W = 143, p = 0.86, Wilcoxon signed-rank test), but the performance is worsened by regressing out the pupil-linked spectral power (W = 241, p = 0.008, Wilcoxon signed-rank test). These results indicate that both arousal-linked and arousal-independent spectral power fluctuations at baseline predict subsequent recognition behavior, but our recordings of pupil size did not provide additional information contributing to the prediction of behavior that was not already present in cortical spectral power.”

This new analysis confirms our main conclusion that both arousal-linked and arousal-independent spectral power fluctuations at baseline predict subsequent recognition behavior.

References:

Aston-Jones G, Cohen JD. 2005. An integrative theory of locus coeruleus-Norepinephrine function: Adaptive Gain and Optimal Performance. Annu Rev Neurosci 28:403–450. doi:10.1146/annurev.neuro.28.061604.135709

Barr DJ, Levy R, Scheepers C, Tily HJ. 2013. Random effects structure for confirmatory hypothesis testing: Keep it maximal. J Mem Lang 68:255–278. doi:10.1016/j.jml.2012.11.001

de Gee JW, Colizoli O, Kloosterman NA, Knapen T, Nieuwenhuis S, Donner TH. 2017. Dynamic modulation of decision biases by brainstem arousal systems. *ELife* 6:e23232. doi:10.7554/*eLife*.23232

Hong L, Walz JM, Sajda P. 2014. Your eyes give you away: Prestimulus changes in pupil diameter correlate with poststimulus task-related EEG dynamics. PLoS One 9:e91321.

doi:10.1371/journal.pone.0091321

Knapen T, De Gee JW, Brascamp J, Nuiten S, Hoppenbrouwers S, Theeuwes J. 2016. Cognitive and ocular factors jointly determine pupil responses under equiluminance. PLoS One 11.

doi:10.1371/journal.pone.0155574

Lendner JD, Helfrich RF, Mander BA, Romundstad L, Lin JJ, Walker MP, Larsson PG, Knight RT. 2020. An electrophysiological marker of arousal level in humans. *ELife* 9:1–29. doi:10.7554/*eLife*.55092

McGinley MJ, David S V., McCormick DA. 2015. Cortical Membrane Potential Signature of Optimal States for Sensory Signal Detection. Neuron 87:179–192. doi:10.1016/j.neuron.2015.05.038

Nakagawa S, Schielzeth H. 2013. A general and simple method for obtaining R2 from generalized linear mixed-effects models. Methods Ecol Evol 4:133–142. doi:10.1111/j.2041-210x.2012.00261.x

Podvalny E, Flounders MW, King LE, Holroyd T, He BJ. 2019. A dual role of prestimulus spontaneous neural activity in visual object recognition. Nat Commun 10:3910. doi:10.1038/s41467-019-11877-4

Powell MJD. 1964. An efficient method for finding the minimum of a function of several variables without calculating derivatives. Comput J 7:155–162. doi:10.1093/comjnl/7.2.155

Steinmetz NA, Zatka-Haas P, Carandini M, Harris KD. 2019. Distributed coding of choice, action and engagement across the mouse brain. Nature 576:266–273. doi:10.1038/s41586-019-1787-x

Van Kempen J, Loughnane GM, Newman DP, Kelly SP, Thiele A, O’Connell RG, Bellgrove MA, O’Connell RG, Bellgrove MA. 2019. Behavioural and neural signatures of perceptual decisionmaking are modulated by pupil-linked arousal. *ELife* 8:e42541. doi:10.7554/*eLife*.42541

Waschke L, Tune S, Obleser J. 2019. Local cortical desynchronization and pupil-linked arousal differentially shape brain states for optimal sensory performance. *ELife* 8. doi:10.7554/*eLife*.51501 Watson AB, Pelli DG. 1983. Quest: A Bayesian adaptive psychometric method. Percept Psychophys 33:113–120. doi:10.3758/BF03202828

[Editors’ note: what follows is the authors’ response to the second round of review.]

Essential revisions:Presentation of single subject data (as per Rev #1)

The individual subject data for the effect of prestimulus pupil size on task performance (Figures 4C-D) was already presented in the previous version of the manuscript in Supplementary figure 3 (Figure 4—figure supplement 1 in the revised version). We now add individual subject data to show the relationship between pupil size and spectral power in Figure 3—figure supplements 2 and 3, for task baseline and rest respectively. Given the large number of data points we made these figures larger and we hope this visual presentation will be helpful to some readers. We also now added individual subject data for decoding accuracy in Figure 4—figure supplement 1.

1) Presentation of effect sizes and explained variance (as per Rev #1).

Marginal and conditional variance explained values are now added to Table 1, Supplementary tables 1 and 2. We used *R’s lme4* through *pymer*, a python wrapper package, to fit the models, and found nearly identical model parameters to stats models:

**Author response image 1. sa2fig1:** 

For this reason, we opted to not transfer all our code to R as it is streamlined in python already. Since our models include more than one random factor in the case of quadratic model, we used *PiecewiseSEM R* package to quantify the marginal and conditional R^2^. The methods section is updated as follow (lines 570-576):“We verified that the parameters estimates acquired with stats models were nearly identical to the ones acquired with a more widely used R lme4^86^ toolbox (implemented through python wrapper pymer^87^). We used piecewiseSEM^88^ R package to estimate marginal and conditional variance-explained values indicating the proportion of total variance explained by fixed effects only and the proportion of variance explained by both fixed and random effects, respectively^89^. It is important to note, however, these metrics were developed specifically for linear mixed-effects models and are useful in the relative sense for comparison between such models’ fit to the data but they do not have the same properties as linear models R^2^. ”

2) Are pupil-related effects really essentially uniform across the cortex? Or are the differences really the important thing? It would be nice if this could be addressed even briefly in the Discussion. (as per Rev #3)

We now added a discussion paragraph addressing this topic (lines 334-343):

“The brain activity modulation by pupil-linked arousal systems, while similar across resting state networks, is not entirely uniform. We observed, for example, differences in parameter estimates across networks and behavioral conditions: modulation of α power by arousal in the visual RSN is stronger than in other networks at rest, whereas modulation of β power is strongest in the dorsal attention network in task baseline (Figure 3C, linear components). Such non-uniform effects are consistent with the intriguing proposals of pupil-linked arousal systems playing an important role in dynamic reconfiguration of large-scale networks topology responding to cognitive demands^50,61^. Indeed, while the arousal neuromodulatory systems send widespread projections to cortex, recent studies also report complex topographic organization which may produce such non-uniform modulatory effects^62,63^. A greater understanding of the function of uniform vs. modular effects on cortical power by arousal systems in various cognitive states is an important goal for future investigation.”

3) Since the present analysis is focused on quantifying pupil size it is important to account for other factors which might affect measured pupil size, notably, fixation and saccades. The section devoted to detailing the analysis of the pupil data omits any discussion of these factors. Did the authors check that fixation is maintained at all times? Were periods associated with saccades away from fixation (which will result in apparent changes to pupil size) removed from the analysis? (as per Rev #4)

The subjects were instructed (and reminded between blocks) to fixate on the center crosshair and to avoid blinking to the best they can until after stimulus arrival (now clarified in lines 495-497) and the experimenters monitored the subjects at all times. The time periods with missing pupil data were removed from the analysis (lines 505-507). These time periods include both blinks and potential saccades, since missing pupil data points is a necessary condition for both saccade and blink detection in EyeLink data (see also our detailed response to reviewer #3, point 5, in previous revision).

4) The analysis of "fast events" was not very clear to us (as per Rev #4, please follow the details for an additional/re-analysis there):a) we are worried about the low-pass filter and a potential acausal smearing it might have contributed.

We used a zero-phase filter – now clarified in the methods section (lines 513-515):

“Next, in order to remove potential high-frequency artifacts, we applied a 5-Hz low-pass Butterworth filter to continued resting state data forward and backward in time using filtfilt function (Scipy), assuring no phase shifts.“

We have repeated the analysis without filtering the pupil signal and the results (now reported in Figure 6-figure supplement 1C) are virtually the same.

b) The statement (line 492) "to be considered a peak/through, the data point should have highest/lowest amplitude within 500 ms vicinity" was also not clear. It would be helpful if you plotted (or reported) the duration distribution of the events you ended up analyzing(as per Rev #4, see details also below).

We have now simplified and clarified the methods for pupillary events identification (lines 516-518):

“To identify peaks/troughs, for each data point in the pupil size time course, we tested whether its amplitude is the highest/lowest in comparison to the data points in a 1-sec time window centered on that data point (using “argrelextrema” method of scipy).”

Because in the present study the events were defined as momentary peaks/troughs, duration of each event is undefined. In contrast to Joshi et al. 2016 where each dilation and constriction event were included in the analysis and a dilation necessarily follows constriction and vice versa, in our analysis constriction and dilations are identified independently, generally allowing for more than one dilation/constriction event in a row and not limiting inter-event interval timing. From the inter-event interval histogram of the events identified in our study (now added as Figure 6—figure supplement 1A) it is possible to conclude that our events could be sparser than those defined in Joshi et al. 2016.

We suspect the reviewers might be interested to know whether the event-related brain activity we report is sensitive to event definition or specifically inter-event interval (duration). To this end, we replicated the analysis with a different method for event identification. This time we used *scipy “findpeaks”* method, where we defined local events by minimal prominence (i.e., peak height in relationship to local minima) of 0.5 standard deviations and selected events with minimal inter-event distance of 0.5, 1, 1.5, 2, 2.5 s. The results (now reported as Figure 6—figure supplement 1B) show virtually no difference from our report (compare to figure 6C)

Thus, we conclude that our reported results are robust to the particular method used to identify these fast events.

Reviewer #1:The authors have revised large parts of their manuscript and addressed most comments.However, two of my previous concerns have not been resolved entirely.1) Presentation of single subject data.The authors argue that this is not feasible as it would require to plot more than 8000 data points. This likely is a misunderstanding. While it is true that plotting single subject points for all power / frequency / ROI bins is not feasible, this is not what I asked for. Take figure 3B as an example. In every panel, the authors plot spectral power (in bins) as a function of pupil size (in bins). They plot one data point per bin (with an error bar around it). If I am not entirely mistaken, this data point represents the group average. It is essential, however, to show single subject data at this point. What speaks against plotting a cloud of N=24 dots around their mean instead of each dot? I realize that this might take up a bit more space but I am convinced that this is an essential service to every reader. Importantly, the same comment applies to figure panels 4C, 4D, and 4E.

We thank the reviewer for the clarification. We now address this comment in essential revisions point #1 above.

2) Effect sizes and explained varianceWhile it is true that the concept of variance is explained is not easily transferred to the world of linear mixed models, it is entirely possible to offer such an estimate, both for linear mixed models as well as generalized mixed models. Even if imperfect, such an approximation represents an important piece of information since it puts the models in perspective to the space they were trying to capture. I allow myself to recommend the excellent R-package lme4 which allows the fitting of such models and in combination with functions from the sj_plot package (e.g., tab_model) offers all relevant information. I am sure transferring their models from python to R should be a simple task for the authors.Furthermore, I suggest to offer an explicit discussion of model fits and effect sizes which is missing in the current version of the manuscript. The supplementary tables illustrate that effects rather are small and likely don't account for large parts of variance in the data. This does not have to be an issue but it should be acknowledged prominently and discussed.

We now address this comment in detail under essential revisions point #2 above. Variances explained (marginal and conditional R^2^) are now reported in Table 1, Supplementary Tables 1 and 2. Conditional R^2^ (i.e., proportion of variance explained by both fixed and random effects) in Table 1 and Table S2 (corresponding to Figures 4C-D and 5B) are in the range of ~0.5–0.9, suggesting that both pupil and residual power explain substantial fractions of variance in the behavioral measures (HR, FAR, c, d’, Accuracy and RT). Conditional R^2^ in Table S1 (corresponding to Figure 3B and Figure 3—figure supplement 1) are much lower. This is expected, because spontaneous cortical power fluctuations have many sources of contributions such as recurrent corticocortical activity, thalamocortical loops, and subcortical neuromodulatory influences. Pupil-linked arousal (part of subcortical neuromodulation) is only one of many sources of influence that contribute to spontaneous cortical power fluctuations.

Reviewer #3:The authors were responsive to all the concerns raised in the initial review. The revised manuscript clarifies the significance of the work and more rigorously identifies the link between pupil size, cortical activity and task performance.

Thank you.

Overall, the manuscript looks great. Just a small lingering question. The effects in Figure 3 appear strikingly consistent across brain networks, and any differences appear to be quantitative more than qualitative (ie, even where a quadratic effect may not be significant, one can still see a trend that is consistent with networks showing a quadratic effect). Does this mean that pupil-related effects are more or less uniform across the cortex? Or are the differences really the important thing? It would be nice if this could be addressed even briefly in the Discussion.

Thank you! We now address this question as part of essential revisions point #3 part above.

Reviewer #4:I joined the review process at the revision stage. I did not read the previous version of the manuscript or the rebuttal later in order to form an unbiased view of the work.1) The experiment is quite long (over an hour and a half). In addition to task related effort, experimental length can have consequences for pupil dynamics due to effort related to maintain fixation and accommodation. It is critical that the authors demonstrate that pre-stimulus pupil size is not correlated with time within the experiment.

The pupil size data was centered within each block, removing any potential correlation between pupil-size and block number (now clarified in Methods, line 511). The subjects were taking breaks of self-paced duration between blocks (line 495) and were encouraged to close their eyes and continue the experiment only as they are ready. Our pupil recording methodology does not allow testing whether the pupil size is correlated to time within experiment because the recording is stopped and restarted between blocks.

2) Similarly, since the present analysis is focused on quantifying pupil size it is important to account for other factors which might affect measured pupil size, notably, fixation and saccades. The section devoted to detailing the analysis of the pupil data omits any discussion of these factors. Did the authors check that fixation is maintained at all times? Were periods associated with saccades away from fixation (which will result in apparent changes to pupil size) removed from the analysis?

We addressed this comment in the essential revisions section #4.

3) The analysis of "fast events" was not very clear to me.a) Why was a low pass filter applied to the data before identifying the phasic events? Wouldn't a low pass filter smear the timing of these events? Depending on the filter (no details are provided) this could result in latency shifts on the order of 200ms, making the "fast event" analysis (Figure 6) difficult to interpret.

We addressed this comment in the essential revisions section #5.

b) The statement (line 492) "to be considered a peak/through, the data point should have highest/lowest amplitude within 500 ms vicinity" was also not clear. It would be helpful if you plotted (or reported) the duration distribution of the events you ended up analyzing and maybe follow the analysis approach as used in e.g. Joshi et al. (2016; Neuron) where phasic events are defined as zero-crossings of the pupil slope separated by {greater than or equal to}75 ms. This will make it easier to relate your findings to the literature on phasic pupil responses.

We addressed this comment in the essential revisions section #5.

4) Analysis in Figure 6 (and page 12).a) Why was a Pearson and not spearman correlation used? The latter is less sensitive to outliers and would provide stronger evidence for the claims the authors wish to make.

We are not sure we understand the reviewer’s concern. We tested for a linear relationship between MEG and pupil size data and therefore chose Pearson correlation. In addition, both MEG signal and pupil size are roughly normally distributed (for an example of pupil size distribution, see Figure 1G), fulfilling the requirement of Pearson correlation.

b) Panel C: I am a bit confused by the example of identified pupil events provided. In the methods you state that you defined dilation/constriction events as having a duration of at most 500 ms. But from eye balling the figure the first dilation event (left most blue dot) appears to last more than a second? (Similarly the right most blue dot in the plot).

We address this comment as part of the essential revision section #5.

c) I would like to see more discussion of the shape of the patter of MEG activity triggered by the pupil events. I found the pattern extremely surprising. e.g. that the sharp peak/trough exactly coincides with 0; is it possible that the preceding peak at ~-400 relates to a previous dilation/constriction event? I would appreciate more detailed discussion of what this all means. Similarly the pattern of correlation across channels looks quite systematic. Did the authors try to source localize this pattern?

We have now expanded discussion on this topic in lines 411-417:

“Pupil size correlates with large-scale MEG activity on a faster timescale as well: first, spontaneous variations in MEG activity precede spontaneous variations in the pupil size with a lag of 400 ms (Figure 6A-B); second, pupillary events (constriction and dilation) coincide with MEG activity peak at lag zero and are preceded by an MEG activity peak occurring 400 ms earlier (Figure 6C-D). A similar pattern of two peaks (at 0 ms and -300 ms) was observed in monkey LC spike rates^5^. The zero-lag peak we observed could result from a neural event occurring earlier in time and influencing both pupil size and MEG activity at the same time lag. Since MEG activity originates mostly in the cortex, the peak at -400 ms lag (100 ms earlier than the peak in monkey LC) could point to an influence of cortical resting state fluctuations on LC activity which then triggers pupillary events.”

We do not think the -400ms peak relates to previous pupillary event since there are no evident rhythms in pupil size fluctuations (Figure 1F). Furthermore, the distribution of inter-event intervals for both dilation and constriction peaks at ~1 sec but with a very long tail (Supplementary figure 6A).

We did not attempt to localize the MEG activity patterns related to pupillary fast events, which we believe is beyond the scope of this paper.

Other points1) It is stated that sensitivity (d') is related to prestimulus size in an inverted U shaped relationship. This is not obvious at all from looking at the data points in the figure. Instead the behavioral performance appears to not differ much for pupil bins 2-5.

Our data is consistent with an inverted-U-shaped relationship suggested by Yerkes-Dodson law. We explain that our study does not explore the whole continuum of arousal states (lines 431-432), which may have limited our ability to capture the full inverted-U-shape function.

2) I understand it is difficult to control in the present experiment, but doesn't a larger pupil also imply stronger visual input? Would this be able to explain the largely linear link between pupil size and brain activity in the β and γ ranges?

Visual input leads to α power suppression, which would be inconsistent with our observation of a positive/inverted-U function between pupil size and α power. Γ power in localized visual areas indeed is known to increase with visual input, however, this phenomenon would not explain the largescale changes in γ power we observed.